# Certifying Robustness to Programmable Data Bias in Decision Trees

**Anna P. Meyer, Aws Albarghouthi,**\* **and Loris D'Antoni**
Department of Computer Sciences
University of Wisconsin–Madison
Madison, WI 53706
{annameyer, aws, loris}@cs.wisc.edu

## Abstract

Datasets can be biased due to societal inequities, human biases, under-representation of minorities, etc. Our goal is to *certify* that models produced by a learning algorithm are *pointwise-robust* to potential dataset biases. This is a challenging problem: it entails learning models for a large, or even infinite, number of datasets, ensuring that they all produce the same prediction. We focus on decision-tree learning due to the interpretable nature of the models. Our approach allows programmatically specifying *bias models* across a variety of dimensions (e.g., missing data for minorities), composing types of bias, and targeting bias towards a specific group. To certify robustness, we use a novel symbolic technique to evaluate a decision-tree learner on a large, or infinite, number of datasets, certifying that each and every dataset produces the same prediction for a specific test point. We evaluate our approach on datasets that are commonly used in the fairness literature, and demonstrate our approach's viability on a range of bias models.

## 1 Introduction

The proliferation of machine-learning algorithms has raised alarming questions about fairness in automated decision-making [4]. In this paper, we focus our attention on bias in training data. Data can be biased due to societal inequities, human biases, under-representation of minorities, malicious data *poisoning*, etc. For instance, historical data can contain human biases, e.g., certain individuals' loan requests get rejected, although (if discrimination were not present) they should have been approved, or women in certain departments are consistently given lower performance scores by managers.

Given biased training data, we are often unable to de-bias it because we do not know which samples are affected. This paper asks, *can we certify (prove) that our predictions are robust under a given form and degree of bias in the training data?* We aim to answer this question without having to show which data are biased (i.e., poisoned). Techniques for certifying poisoning robustness (*i*) focus on specific poisoning forms, e.g., label-flipping [31], or (*ii*) perform certification using defenses that create complex, uninterpretable classifiers, e.g., due to randomization or ensembling [23, 24, 31]. To address limitation (*i*), we present *programmable bias definitions* that model nuanced biases in practical domains. To address (*ii*), we target *existing* decision-tree learners—considered interpretable and desirable for sensitive decision-making [32]—and exactly certify their robustness, i.e., provide proofs that the bias in the data will not affect the outcome of the trained model on a given point.

We begin by presenting a *language* for programmatically defining *bias models*. A bias model allows us to flexibly specify what sort of bias we suspect to be in the data, e.g., up to $n\%$ of the women *may have* wrongly received a negative job evaluation. Our bias-model language is generic, allowing

---

\* Author's name in native alphabet: أوس البرغوثي

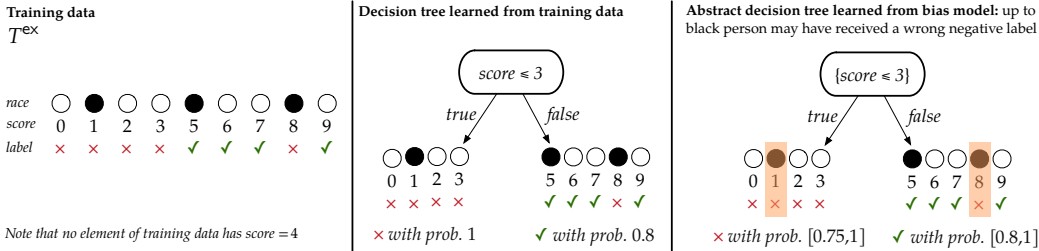

Figure 1: A simple, hypothetical running example

us to *compose* simpler bias models into more complex ones, e.g., up to $n\%$ of the women may have wrongly received a negative evaluation *and* up to $m\%$ of Black men's records may have been completely missed. The choice of bias model depends on the provenance of the data and the task.

After specifying a bias model, our goal is to certify *pointwise robustness to data bias*: Given an input $x$, we want to ensure that no matter whether the training data is biased or not, the resulting model's prediction for $x$ remains the same. Certifying pointwise robustness is challenging. One can train a model for every perturbation (as per a bias model) of a dataset and make sure they all agree. But this is generally not feasible, because the set of possible perturbations can be large or infinite. Recall the bias model where up to $n\%$ of women may have wrongly received a negative label. For a dataset with 1000 women and $n = 1\%$, there are more than $10^{23}$ possible perturbed datasets.

To perform bias-robustness certification on decision-tree learners, we employ *abstract interpretation* [12] to symbolically run the decision-tree-learning algorithm on a large or infinite set of datasets simultaneously, thus learning a *set* of possible decision trees, represented compactly. The crux of our approach is a technique that lifts operations of decision-tree learning to symbolically operate over a *set of datasets* defined using our bias-model language. As a starting point, we build upon Drews et al.'s [16] demonstration of poisoning-robustness certification for the simple bias model where an adversary may have added fake training data. Our approach completely reworks and extends their technique to target the bias-robustness problem and handle complex bias models, including ones that may result in an infinite number of datasets.

**Contributions.** We make three contributions: (1) We formalize the bias-robustness-certification problem and present a language to compositionally define bias models. (2) We present a symbolic technique that performs decision-tree learning on a set of datasets defined by a bias model, allowing us to perform certification. (3) We evaluate our approach on a number of bias models and datasets from the fairness literature. Our tool can certify pointwise robustness for a variety of bias models; we also show that some datasets have *unequal robustness-certification rates* across demographics groups.

**Running example.** Consider the example in Fig. 1; our goal is to classify who should be hired based on a test score. A standard decision-tree-learning algorithm would choose the split (predicate) $score \leqslant 3$, assuming we restrict tree depth to 1.[2] As shown in Fig. 1 (middle), the classification depends on the data split; e.g., on the right hand side, we see that a person with $score > 3$ is accepted, because the proportion ("probability") of the data with positive labels and $score > 3$ is $4/5$ ($> 1/2$).

Now suppose that our bias model says that up to one Black person in the dataset may have received a wrongful rejection. Our goal is to show that even if that is the case, the prediction of a new test sample $x$ will not change. As described above, training decision trees for all possible modified datasets is generally intractable. Instead, we symbolically learn a set of possible decision trees compactly, as illustrated in Fig. 1 (right). In this case the learning algorithm always chooses $score \leqslant 3$ (generally, our algorithm can capture all viable splits). However, the proportion of labels on either branch varies. For example, on the right, if the highlighted sample is wrongly labeled, then the ratio changes from $0.8$ to $1$. To efficiently perform this calculation, we lift the learning algorithm's operations to *interval arithmetic* and represent the probability as $[0.8, 1]$. Given a new test sample $x = \langle race{=}\text{Black},\ score{=}7 \rangle$, we follow the right branch and, since the interval is always larger than $0.5$, we certify that

---

[2]Other predicates, e.g., $score \leqslant 4$, will yield the same split. We choose a single split for illustrative purposes here. (The implementation considers all possible splits that yield distinct partitions, so it would consider $score \leqslant 3$ and $score \leqslant 4$ as a single entity.)

the algorithm is robust for $x$. In general, however, due to the use of abstraction, our approach may fail to find tight intervals, and therefore be unable to certify robustness for all robust inputs.

## 2    Related work

**Ties to poisoning.** Our dataset bias language captures existing definitions of *data poisoning*, where an attacker is assumed to have maliciously modified training data. Poisoning has been studied extensively. Most works have focused on attacks [6, 11, 25, 33, 38, 39, 40] or on training models that are empirically less vulnerable (defenses) [3, 9, 19, 28, 31, 36]. Our work differs along a number of dimensions: (1) We allow programmatic, custom, composable definitions of bias models; notably, to our knowledge, no other work in this space allows for *targeted* bias, i.e., restricting bias to a particular subgroup. (2) Our work aims to certify and quantify robustness of an existing decision-tree algorithm, not to modify it (e.g., via bagging or randomized smoothing) to improve robustness [23, 24, 31].

Statistical defenses show that a learner is robust *with high probability*, often by modifying a base learner using, e.g., randomized smoothing [31], outlier detection [36], or bagging [22, 23]. Non-statistical certification (including abstract interpretation) has mainly focused on *test-time* robustness, where the vicinity (e.g., within an $\ell_p$ norm) of an input is proved to receive the same prediction [2, 21, 30, 35, 37, 1]. Test-time robustness is a simpler problem than our *train-time* robustness problem because it does not have to consider the mechanics of the learner on sets of datasets. The only work we know of that certifies train-time robustness of decision trees is by Drews et al. [16] and focuses on poisoning attacks where an adversary *adds fake data*. Our work makes a number of significant leaps beyond this work: (1) We frame data bias as programmable, rather than fixed, to mimic real-world bias scenarios, an idea that has gained traction in a variety of domains, e.g., NLP [41]. (2) We lift a decision-tree-learning algorithm to operate over sets of datasets represented via our bias-model language. (3) We investigate the bias-robustness problem through a fairness lens, particularly with an eye towards robustness rates for various demographic groups.

**Ties to fairness.** The notion of *individual fairness* specifies that *similar individuals* should receive similar predictions [18]; by contrast, we certify that no individual should receive different predictions under models trained by *similar datasets*. Black and Fredrickson explore the problem of how individuals' predictions change under models trained by similar datasets, but their concept of similarity is limited to removing a *single* data point [7]. Data bias, in particular, has received some attention in the fairness literature. Chen et al. suggest adding missing data as an effective approach to remedying bias in machine learning [10], which is one operation that our bias language captures. Mandal et al. build on the field of *distributional robustness* [5, 27, 34] to build classifiers that are empirically group-fair across a variety of nearby distributions [26]. Our problem domain is related to distributional robustness because we certify robustness over a family of similar datasets; however, we define specific data-transformation operators to define similarity, and, unlike Mandal et al., we certify existing learners instead of building empirically robust models.

**Ties to robust statistics.** There has been renewed interest in robust statistics for machine learning [13, 14]. Much of the work concerns outlier detection for various learning settings, e.g., estimating parameters of a Gaussian. The distinctions are two-fold: (1) We deal with rich, nuanced bias models, as opposed to out-of-distribution samples, and (2) we aim to certify that predictions are robust for a specific input, a guarantee that cannot be made by robust-statistics-based techniques [15].

## 3    Defining data bias programmatically

We define the *bias-robustness problem* and a language for defining bias models *programmatically*.

**Bias models.** A dataset $T \subseteq \mathcal{X} \times \mathcal{Y}$ is a set of pairs of samples and labels, where $\mathcal{Y} = \{0, \dots, n-1\}$. For a dataset $T$, we will use $T_{\mathcal{X}}$ to denote $\{x \mid (x, y) \in T\}$. A *bias model* $B$ is a function that takes a dataset and returns a *set of datasets*. We call $B(T) \subseteq 2^{\mathcal{X} \times \mathcal{Y}}$ a *bias set*. We assume that $T \in B(T)$. Intuitively, $B(T)$ represents *all datasets that could have existed had there been no bias*.

**Pointwise data-bias robustness.** Assume we have a learning algorithm $A$ that, given a training dataset $T$, deterministically returns a classifier $h_T$ from some hypothesis class $\mathcal{H}$. Fix a dataset $T$ and bias model $B$. Given a sample $x \in \mathcal{X}$, we say that $A$ is *pointwise robust* (or robust for short) on $x$ iff

$$\text{there is a label } i \text{ such that for all } T' \in B(T), \text{ we have } h_{T'}(x) = i \tag{1}$$

**Basic components of a bias model.** We begin with basic bias models.

*Missing data*: A common bias in datasets is missing data, which can occur via poor historical representation of a subgroup (e.g., women in CS-department admissions data), or from present-day biases or shortsightedness (e.g., a survey that bypasses low-income neighborhoods). Using the parameter $m$ as the maximum number of missing elements, we formally define:

$$\text{MISS}_m(T) = \{T' \mid T' \supseteq T,\ |T' \backslash T| \leqslant m\}$$

$\text{MISS}_m(T)$ defines an *infinite* number datasets when the sample space is infinite (e.g., $\mathbb{R}$-valued).

**Example 3.1.** *Using $T^{\text{ex}}$ from Fig. 1, $\text{MISS}_1(T^{\text{ex}})$ is the set of all datasets that are either $T^{\text{ex}}$ or $T^{\text{ex}}$ plus any new element $(x, y)$ with arbitrary race, score, and label.*

*Label flipping*: Historical data can contain human biases, e.g., in loan financing, certain individuals' loan requests get rejected due to discrimination. Or consider employee-performance data, where women in certain departments are consistently given lower scores by managers. We model such biases as label flipping, where labels of up to $l$ individuals in the dataset may be incorrect:

$$\text{FLIP}_l(T) = \{T' \mid |T|{=}|T'|,\ |T \backslash T'| \leqslant l,\ T_{\mathcal{X}} = T'_{\mathcal{X}}\}$$

**Example 3.2.** *Using $T^{\text{ex}}$ from Fig. 1 and a bias model $\text{FLIP}_1$, we have $\text{FLIP}_1(T^{\text{ex}}) = \{T_0^{\text{ex}}, \ldots, T_3^{\text{ex}}, T_5^{\text{ex}}, \ldots, T_9^{\text{ex}}, T^{\text{ex}}\}$, where $T_i^{\text{ex}}$ is $T^{\text{ex}}$ with the label of the element with $\text{score} = i$ changed.*

*Fake data*: Our final bias model assumes the dataset may contain *fake* data. One cause may be a malicious user who enters fraudulent data into a system (often referred to as *poisoning*). Alternatively, this model can be thought of as the inverse of $\text{MISS}$, e.g., we over-collected data for men.

$$\text{FAKE}_k(T) = \{T' \mid T' \subseteq T,\ |T \backslash T'| \leqslant k\}$$

**Example 3.3.** *Using $T^{\text{ex}}$ from Fig. 1 and a bias model $\text{FAKE}_1$, we get $\text{FAKE}_1(T^{\text{ex}}) = \{T_{\downarrow 0}^{\text{ex}}, \ldots, T_{\downarrow 3}^{\text{ex}}, T_{\downarrow 5}^{\text{ex}}, \ldots, T_{\downarrow 9}^{\text{ex}}, T^{\text{ex}}\}$ where $T_{\downarrow i}^{\text{ex}}$ is $T^{\text{ex}}$ such that the element with $\text{score} = i$ has been removed.*

**Targeted bias models.** Each bias model has a *targeted* version that limits the bias to a specified group of data points. For example, consider the missing data transformation. If we suspect that data about women is missing from an HR database, we can limit the $\text{MISS}$ transformation to only add data points with $\text{gender} = \text{female}$. Formally, we define a predicate $g : \mathcal{X} \times \mathcal{Y} \to \mathbb{B}$, where $\mathbb{B} = \{\text{true},\ \text{false}\}$.

$$\text{MISS}_m^g(T) = \{T' \mid T' \supseteq T,\ |T' \backslash T| \leqslant m,\ \text{and } g(x, y) \text{ is true } \forall (x, y) \in (T' \backslash T)\}$$

Targeted versions of label-flipping and fake data can be defined in a similar way.

**Example 3.4.** *In Fig. 1 (right), we used bias model $\text{FLIP}_1^g$, where $g$ targets Black people with negative labels. This results in the bias set $\text{FLIP}_1^g(T^{\text{ex}}) = \{T_1^{\text{ex}}, T_8^{\text{ex}}, T^{\text{ex}}\}$, where $T_i^{\text{ex}}$ is $T^{\text{ex}}$ with the label of the element with $\text{score} = i$ changed (recall that scores 1 and 8 belong to Black people in $T^{\text{ex}}$).*

**Composite bias models.** We can compose basic components to generate a *composite model*. Specifically, we define a composite model $B$ as a finite set of arbitrary basic components, that is,

$$B = [\text{MISS}_{m_1}^{g_1}, \ldots, \text{MISS}_{m_j}^{g_j}, \text{FLIP}_{l_1}^{g_{j+1}}, \ldots, \text{FLIP}_{l_p}^{g_{j+p}}, \text{FAKE}_{k_1}^{g_{j+p+1}}, \ldots, \text{FAKE}_{k_q}^{g_{j+p+q}}] \qquad (2)$$

$B(T)$ is generated from $T$ by applying the basic components of $B$ iteratively. We must apply the constituent components in an optimal order, i.e., one that generates all datasets that can be created through applying the transformers in *any* order. To do this, we apply components of the same type in any order and apply transformers of different types in the order $\text{MISS}$, $\text{FLIP}$, $\text{FAKE}$ (see Appendix).

**Example 3.5.** *Suppose $B = [\text{MISS}_2^{g_1}, \text{FAKE}_1]$. Then $B(T)$ is the set of all datasets obtained by adding up to 2 arbitrary data points that satisfy $g_1$ to $T$, and then removing any up to 1 data point.*

## 4  Certifying robustness for decision-tree learning

We begin with a simplified version of the CART algorithm [8], which is our target for certification.

Given a dataset $T$ and a Boolean function (predicate) $\phi : \mathcal{X} \to \mathbb{B}$, we define:

$$T_\phi = \{(x, y) \in T \mid \phi(x)\},$$

i.e., $T_\phi$ is the set of elements satisfying $\phi$. Analogously, $T_{\neg \phi} = \{(x, y) \in T \mid \neg \phi(x)\}$.

**Example 4.1.** *Using $\phi \triangleq score \leqslant 3$, we have $T_\phi^{\text{ex}} = \{0, 1, 2, 3\}$ and $T_{\neg\phi}^{\text{ex}} = \{5, 6, 7, 8, 9\}$.*

**Learning algorithm.** To formalize our approach, it suffices to consider a simple algorithm that learns a decision stump, i.e., a tree of depth 1. Therefore, the job of the algorithm is to choose a predicate (splitting rule) $\phi$ from a set of predicates $\Phi$ that optimally splits the dataset $T$ into two datasets. Formally, we define $\text{pr}_i(T)$ as the proportion of $T$ with label $i$, i.e.,

$$\text{pr}_i(T) = |\{(x, i) \in T\}| \, / \, |T| \tag{3}$$

We use pr to calculate *Gini impurity* (imp), that is,

$$\text{imp}(T) = \sum_{i=0}^{n-1} \text{pr}_i(T)(1 - \text{pr}_i(T))$$

Using imp, we assign each dataset-predicate pair a cost, where a low value indicates that $\phi$ splits $T$ *cleanly*, i.e., elements of $T_\phi$ (conversely, $T_{\neg\phi}$) have mostly the same label:

$$\text{cost}(T, \phi) = |T_\phi| \cdot \text{imp}(T_\phi) + |T_{\neg\phi}| \cdot \text{imp}(T_{\neg\phi})$$

Finally, we select the predicate that results in the lowest cost (we break ties arbitrarily), as defined by the split operator:

$$\text{split}(T) = \operatorname*{argmin}_{\phi \in \Phi} \text{cost}(T, \phi)$$

**Example 4.2.** *For $\phi \triangleq score \leqslant 3$, $\text{cost}(T^{\text{ex}}, \phi) = 4 \times 0 + 5 \times 0.32 = 1.6$.*

**Inference.** Given an optimal predicate $\phi$ and a new sample $x$ to classify, we return the label with the highest proportion in the branch of the tree that $x$ takes. Formally,

$$\text{infer}(T, \phi, x) = \operatorname*{argmax}_i \text{pr}_i(T_{\phi'}),$$

where $\phi'$ is $\phi$ if $\phi(x) = true$; otherwise, $\phi'$ is $\neg\phi$.

### 4.1 Certifying bias robustness with abstraction

Given a dataset $T$, bias model $B$, and sample $x$, our goal is to prove robustness (Eq. (1)): no matter which dataset in $B(T)$ was used to learn a decision tree, the predicted label of $x$ is the same. Formally,

$$\text{there is a label } i \text{ s.t. for all } T' \in B(T), \text{infer}(T', \phi', x) = i, \text{ where } \phi' = \text{split}(T') \tag{4}$$

The naïve way to prove this is to learn a decision tree using each dataset in $B(T)$ and compare the results. This approach is intractable or impossible, as $|B(T)|$ may be combinatorially large or infinite.

Instead, we *abstractly evaluate* the decision-tree-learning algorithm on the entire bias set $B(T)$ in a symbolic fashion, without having to enumerate all datasets. Specifically, for each operator in the decision-tree-learning algorithm, we define an abstract analogue, called an *abstract transformer* [12], that operates over *sets of training sets* symbolically. An abstract transformer is an approximation of the original operator, in that it *over-approximates* the set of possible outputs on the set $B(T)$.

**Sound abstract transformers.** Consider the $\text{pr}_i$ operator, which takes a dataset and returns a real number. We define an abstract transformer $\text{pr}_i^a(B(T))$ that takes a set of datasets (defined as a bias set) and returns an interval, i.e., a *subset* of $\mathbb{R}$. The resulting interval defines a range of possible values for the probability of class $i$. E.g., an interval $\text{pr}_i^a(B(T))$ may be $[0.1, 0.3]$, meaning that the proportion of $i$-labeled elements in datasets in $B(T)$ is between $0.1$ and $0.3$, inclusive.

Given intervals computed by $\text{pr}_i^a$, downstream operators will be lifted into interval arithmetic, which is fairly standard. E.g., for $a, b, c, d \in \mathbb{R}$, $[a, b] + [c, d] = [a + c, b + d]$. It will be clear from context when we are applying arithmetic operators to intervals. For a sequence $\{x_i\}_i$ of intervals, $\operatorname{argmax}_i x_i$ returns a *set of possible indices*, as intervals may overlap and there may be no unique maximum.

**Example 4.3.** *Let $I = \{[1, 2], [4, 8], [6, 7], [4, 5]\}$. Then $\max(I) = \{[4, 8], [6, 7]\}$ because 6 is the greatest lower bound of $I$, and $[4, 8]$ and $[6, 7]$ are the only intervals in $I$ that contain 6.*

For the entire certification procedure to be correct, $\text{pr}^a$ and all other abstract transformers must be *sound*. That is, they should over-approximate the set of possible outputs. Formally, $\text{pr}_i^a$ is a sound approximation of $\text{pr}_i$ iff for all $i$ and all $T' \in B(T)$, we have $\text{pr}_i(T') \in \text{pr}_i^a(B(T))$.

**Certification process.** To perform certification, we use an abstract transformer $\text{split}^a(B(T))$ to compute a *set* of best predicates $\Phi^a$ for $B(T)$. The reason $\text{split}^a$ returns a set of predicates is because its input is a set of datasets that may result in different optimal splits. Then, we use an abstract transformer $\text{infer}^a(B(T), \Phi^a, x)$ to compute a *set* of labels for $x$. If $\text{infer}^a$ returns a singleton set, then we have proven pointwise robustness for $x$ (Eq. (4)); otherwise, we have an inconclusive result—we cannot falsify robustness because abstract transformers are over-approximate.

## 4.2 Abstract transformers for $\text{pr}$

We focus on the most challenging transformer, $\text{pr}^a$; in § 4.3, we show the rest of the transformers.

**Abstracting missing data.** We begin by describing $\text{pr}^a$ for missing data bias, $B = \text{MISS}_m$. From now on, we use $c_i$ to denote the number of samples $(x, y) \in T$ with $y = i$. We define $\text{pr}_i^a$ by considering how we can add data to *minimize* the fraction of $i$'s in $T$ for the lower bound of the interval, and *maximize* the fraction of $i$'s in $T$ for the upper bound. To minimize the fraction of $i$'s, we add $m$ elements with label $j \neq i$; to maximize the fraction of $i$'s, we add $m$ elements with label $i$.

$$\text{pr}_i^a(\text{MISS}_m(T)) = \left[ \frac{c_i}{|T| + m} , \frac{c_i + m}{|T| + m} \right] \tag{5}$$

**Example 4.4.** *Given $B = \text{MISS}_1$, we have* $\text{pr}_{\checkmark}^a(B(T^{\text{ex}})) = \left[ \frac{4}{10}, \frac{6}{10} \right]$.

**Abstracting label-flipping.** Next, we define $\text{pr}_i^a$ for label-flipping bias, where $B = \text{FLIP}_l$. Intuitively, we can minimize the proportion of $i$'s by flipping $l$ labels from $i$ to $j \neq i$, and maximize the proportion of $i$'s by flipping $l$ labels from $j \neq i$ to $i$. The caveat here is that if there are fewer than $l$ of whichever label we want to flip, we are limited by $c_i$ or $\sum_{j \neq i} c_j$, depending on flipping direction.

$$\text{pr}_i^a(\text{FLIP}_l(T)) = \left[ \frac{c_i - \min(l, c_i)}{|T|}, \frac{c_i + \min(l, \sum_{j \neq i} c_j)}{|T|} \right] \tag{6}$$

**Example 4.5.** *Given $B = \text{FLIP}_1$, we have* $\text{pr}_{\checkmark}^a(B(T^{\text{ex}})) = \left[ \frac{4}{9}, \frac{6}{9} \right]$.

Fake data bias models can be abstracted similarly (see Appendix).

**Abstracting targeted bias models.** We now show how to abstract targeted bias models, where a function $g$ restricts the affected samples. To begin, we limit $g$ to only condition on features, not the label. In the case of $\text{MISS}^g$, the definition of $\text{pr}^a$ does not change, because even if we restrict the characteristics of the elements that we can add, we can still add up to $m$ elements with any label.

In the case of label-flipping, we constrain the parameter $l$ to be no larger than $|T_g|$. Formally, we define $l_i = \min(l, |\{(x, i) \in T : g(x)\}|)$ and then

$$\text{pr}_i^a(\text{FLIP}_l^g(T)) = \left[ \frac{c_i - l_i}{|T|}, \frac{c_i + \min(\sum_{j \neq i} l_j, l)}{|T|} \right] \tag{7}$$

The definition for fake data is similar (see Appendix). The above definition is sound when $g$ conditions on the label; however, the Appendix includes a more precise definition of $\text{pr}^a$ for that scenario.

**Abstracting composite bias models.** Now consider a composite bias model consisting of all the basic bias models. Intuitively, $\text{pr}_i^a$ will need to reflect changes in $c_i$ that occur from adding data, flipping labels, and removing data. First, we consider a bias model with just one instance of each $\text{MISS}$, $\text{FLIP}$, and $\text{FAKE}$, i.e., $B = [\text{MISS}_m^{g_1}, \text{FLIP}_l^{g_2}, \text{FAKE}_k^{g_3}]$. We define auxiliary variables $l_i = \min(l, |\{(x, i) \in T : g_2(x, i)\}|)$ and $k_i = \min(k, |\{(x, i) \in T : g_3(x, i)\}|)$. Intuitively, these variables represent the number of elements with label $i$ that we can alter. Conversely, to represent the elements with a label other than $i$, we will use $l_i' = \min(l, \Sigma_{j \neq i} l_j)$ and $k_i' = \min(k, \Sigma_{j \neq i} c_j)$.

$$\text{pr}_i^a(B(T)) = \left[ \max\left( 0, \frac{c_i - l_i - k_i}{|T| - k_i + m} \right), \min\left( 1, \frac{c_i + l_i' + m}{|T| - k_i' + m} \right) \right] \tag{8}$$

Extending the above definition to allow multiple uses of the same basic model, e.g., $\{\text{FLIP}_{l_1}^{g_1}, \text{FLIP}_{l_2}^{g_2}\}$ is simple: essentially, we just sum $l_1$ and $l_2$. A full formal definition is in the Appendix.

**Theorem 1.** $\mathsf{pr}^a$ *is a sound abstract transformer. (In the Appendix, we also show that* $\mathsf{pr}^a$ *is precise.)*

### 4.3 An abstract decision-tree algorithm

We define the remaining abstract transformers, with the goal of certification. Our definitions are based on Drews et al. [16]; the key difference is the $T_\phi$ operation, which is dependent on the bias model.

**Filtering.** We need $B(T)_\phi$, the abstract analogue of $T_\phi$. For FLIP and FAKE, we define $B(T)_\phi = B(T_\phi)$. But for MISS, we have to alter the bias model, too, since after filtering on $\phi$ we only want to add new elements that satisfy $\phi$. We define $\mathsf{MISS}_m^g(T)_\phi = \mathsf{MISS}_m^{g\wedge\phi}(T_\phi)$. Filtering composite bias models applies these definitions piece-wise (see a full definition and soundness proof in the Appendix).

**Gini impurity.** We lift imp to interval arithmetic: $\mathsf{imp}^a(T) = \sum_{i=1}^n \mathsf{pr}_i^a(T)([1,1] - \mathsf{pr}_i^a(T))$.

**Cost.** Recall that cost relies on $|T_\phi|$. We want an abstract analogue of $|T_\phi|$ that represents the range of sizes of datasets in $B(T)_\phi$ and not the number of datasets in $B(T)_\phi$. To this end, we define an auxiliary function size where $\mathsf{size}(B(T)_\phi) = [a,b]$ such that $a = \min\{|T'| : T' \in B(T)_\phi\}$ and $b = \max\{|T'| : T' \in B(T)_\phi\}$.

Then, we define the cost of splitting on $\phi$ as follows (recall that the operators use interval arithmetic):

$$\mathsf{cost}^a(B(T), \phi) = \mathsf{size}(B(T)_\phi) \times \mathsf{imp}^a(B(T)_\phi) + \mathsf{size}(B(T)_{\neg\phi}) \times \mathsf{imp}^a(B(T)_{\neg\phi}) \qquad (9)$$

Since size and $\mathsf{imp}^a$ return intervals, $\mathsf{cost}^a$ will be an interval, as well.

**Best split.** To find the set of best predicates, we identify the *least upper bound* ($lub$) of any predicate's cost. Then, any predicate whose cost overlaps with $lub$ will be a member of the set of best predicates, too. Formally, $lub = \min_{\phi\in\Phi} b_\phi$, where $\mathsf{cost}^a(\phi) = [a_\phi, b_\phi]$ Then, we define $\mathsf{split}^a(B(T)) = \{\phi \in \Phi \mid a_\phi \leqslant lub\}$.

**Inference.** Finally, for inference, we evaluate every predicate computed by $\mathsf{split}^a$ on $x$ and collect all possible prediction labels. Intuitively, we break the problem into two pieces: first, we evaluate all predicates $\phi$ that satisfy $\phi(x)$ (i.e., when $x$ is sent down the left branch of the tree), and then predicates that satisfy $\neg\phi(x)$, (i.e., when $x$ is sent down the right branch of the tree). Formally, we compute:

$$\mathsf{infer}^a(B(T), \Phi^a, x) = \underbrace{\bigcup_{\phi(x)} \operatorname*{argmax}_i \mathsf{pr}_i^a(B(T)_\phi)}_{\text{labels for predicates } \phi \text{ s.t. } \phi(x)} \cup \underbrace{\bigcup_{\neg\phi(x)} \operatorname*{argmax}_i \mathsf{pr}_i^a(B(T)_{\neg\phi})}_{\text{labels for predicates } \phi \text{ s.t. } \neg\phi(x)} \qquad (10)$$

where the range of $\cup$ is over predicates in $\Phi^a$. Since our goal is to prove robustness, we only care whether $|\mathsf{infer}^a(B(T), \Phi^a, x)| = 1$, i.e., all datasets produce the same prediction.

**Theorem 2.** *If* $|\mathsf{infer}^a(B(T), \Phi^a, x)| = 1$, *where* $\Phi^a = \mathsf{split}^a(B(T))$, *then* $x$ *is robust (Eq. (4)).*

**Example 4.6.** *Recall Fig. 1 with bias model* $B = \mathsf{FLIP}_1^g$, *where* $g$ *targets Black people with* $\times$ *label.* $\mathsf{split}^a(B(T^{\mathsf{ex}}))$ *returns the singleton set* $\Phi^a = \{score \leqslant 3\}$. *Then, given input* $x = \langle race = Black, score = 7\rangle$, $\mathsf{infer}^a(B(T^{\mathsf{ex}}), \Phi^a, x) = \{\checkmark\}$, *since* $\mathsf{pr}_{\checkmark}^a(B(T^{\mathsf{ex}})_{score>3}) = [0.8, 1]$, *which is greater than* $\mathsf{pr}_{\times}^a(B(T^{\mathsf{ex}})_{score>3}) = [0, 0.2]$. *Therefore, the learner is robust on* $x$.

## 5 Experimental evaluation

We implement our certification technique in C++ and call it Antidote-P, as it extends Antidote [16] to programmable bias models. To learn trees with depth $> 1$, we apply the presented procedure recursively. We use Antidote's *disjunctive domain*, which is beneficial for certification [16] but requires a large amount of memory because it keeps track of many different datasets on each decision-tree path. We evaluate on Adult Income [17] (training $n$=32,561), COMPAS [29] ($n$=4629), and Drug Consumption [20] ($n$=1262). A fourth dataset, MNIST 1/7 ($n$=13,007), is in the Appendix. For all datasets, we use the standard train/test split if one is provided; otherwise, we create our own train/test splits, which are available in our code repository at `https://github.com/annapmeyer/antidote-P`.

Table 1: Certification rates for various bias models. Targeted bias models use predicates (*race* = Black and *label* = positive) for COMPAS and (*gender* = female and *label* = negative) for Adult Income. Composite models show cumulative bias, e.g., 0.2% MISS + FAKE bias equates to 0.1% bias of each MISS and FAKE. Empty entries indicate tests that failed due to memory constraints (96GB).

| | | Bias amount as a percentage of training set | | | | | |
|---|---|---|---|---|---|---|---|
| **Bias type** | **Dataset** | 0.05 | 0.1 | 0.2 | 0.4 | 0.7 | 1.0 |
| MISS (missing data) | Drug Consumption | 94.5 | 94.5 | 94.5 | 94.5 | 85.1 | 85.1 |
| | COMPAS | 89.0 | 81.9 | 52.9 | 45.3 | 9.3 | 9.2 |
| | Adult Income (AI) | 96.0 | 86.9 | 72.8 | 60.9 | | |
| | COMPAS targeted | 89.0 | 89.0 | 81.9 | 52.9 | 47.8 | 42.3 |
| | AI targeted | 98.8 | 97.2 | 86.6 | 73.0 | 62.0 | 31.6 |
| FLIP (label-flipping) | Drug Consumption | 94.5 | 94.5 | 94.5 | 92.1 | 85.1 | 7.1 |
| | COMPAS | 81.9 | 71.5 | 47.8 | 20.6 | 3.0 | 3.0 |
| | Adult Income | 95.8 | 72.9 | 70.2 | 34.8 | | |
| | COMPAS targeted | 89.0 | 81.9 | 71.5 | 50.5 | 43.2 | 24.2 |
| | AI targeted | 98.6 | 97.0 | 74.4 | 71.0 | 45.4 | 25.8 |
| MISS + FAKE (missing + fake) | Drug Consumption | 94.5 | 94.5 | 94.5 | 94.5 | 85.1 | 85.1 |
| | COMPAS | 81.9 | 76.2 | 52.9 | 43.2 | 9.3 | 9.3 |
| | Adult Income | 96.0 | 95.6 | 72.8 | 68.3 | 36.2 | |
| MISS + FLIP (missing + label-flipping) | Drug Consumption | 94.5 | 94.5 | 92.1 | 92.1 | 85.1 | 38.0 |
| | COMPAS | 81.9 | 71.5 | 50.5 | 41.6 | 9.3 | 3.0 |
| | Adult Income | 95.9 | 74.3 | 71.1 | 49.0 | | |
| Bias-set size color scheme | | $< 10^{10}$ | $< 10^{50}$ | $< 10^{100}$ | $< 10^{500}$ | $> 10^{500}$ | infinite |

For each dataset, we choose the smallest tree depth where accuracy improves no more than 1% at the next-highest depth. For Adult Income and MNIST 1/7, this threshold is depth 2 (accuracy 83% and 97%, respectively); for COMPAS and Drug Consumption it is depth 1 (accuracy 64% and 76%, respectively). We run additional experiments on COMPAS and Drug Consumption at depths 2 and 3 to evaluate how tree depth influences Antidote-P's efficiency (see Appendix).

A natural baseline is enumerating all datasets in the bias set but that is infeasible—see bias-set sizes in Table 1. To our knowledge, our technique (extended from [16]), is the only method to certify bias robustness of decision-tree learners.

## 5.1 Effectiveness at certifying robustness

Table 1 shows the results. Each entry in the table indicates the percentage of test samples for which Antidote-P can prove robustness with a given bias model and the shading indicates the size of the bias set, $|B(T)|$. We see that even though the perturbation sets are very large—sometimes infinite—we are able to certify robustness for a significant percentage of elements.

**By dataset.** Certification rates vary from 100% robustness for $\text{MISS}_{0.1\%}$ on Adult Income (i.e., the predictions of each point in the test set does not change if up to 0.1% new points are added to the training set) to just 3% robustness for $\text{FLIP}_{1\%}$ on COMPAS. Even for a single bias model, the certification rates vary widely: under $\text{FLIP}_{0.2\%}$, we can verify 94.5% of samples as robust for Drug Consumption, but only 70.2% for Adult Income and 47.8% for COMPAS. We posit that these differences stem from inherent properties of the datasets. The normalized cost of the optimal top-most split is 0.30 for Adult Income, 0.35 for Drug Consumption, and 0.45 for COMPAS (recall that a lower cost corresponds to greater information gain). As a result, biasing a fixed percentage of data yields greater instability for COMPAS, since the data already exhibited poorer separation.

**By bias model.** There are also differences in certification rates between bias models. FLIP is more destructive to robustness because flipping a single label results in a symmetric difference of 2 from the original dataset (as if we removed an element from the set and then inserted a new one with a flipped label), while adding a single item results in a symmetric difference of 1.

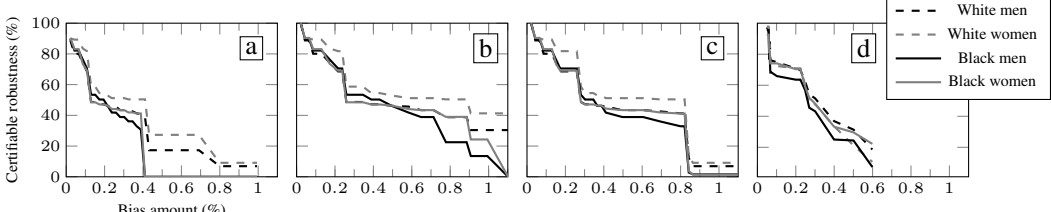

Figure 2: Left to right: Certifiable robustness by demographic group on (a) COMPAS under FLIP; (b) COMPAS under $\text{FLIP}^g$ where $g \triangleq (race = \text{Black} \wedge label = \text{positive})$; (c) COMPAS under $\text{FLIP}^g$ where $g \triangleq (race = \text{White} \wedge label = \text{negative})$; (d) Adult Income under FLIP.

The composite bias models display similar dataset- and bias model-based trends. Notably, MISS + FAKE yields a lower certifiable-robustness rate than FLIP. E.g., under $\text{FLIP}_{0.1\%}$, we can certify 71.5% of COMPAS test samples as robust. But for $\text{MISS}_{0.1\%} + \text{FAKE}_{0.1\%}$ (that is, 0.2% bias total), we are only able to certify 50.5% of test samples as robust. This shows that FLIP is a useful modeling tool for situations where we believe the features of all data points to be trustworthy, but suspect that some labels may be incorrect. The targeted bias models allow for greater certification rates than the non-targeted versions; this is expected because they result in smaller bias sets.

In summary, Antidote-P **can effectively certify robustness across a variety of bias models, but its success depends on properties of the dataset such as separability**.

## 5.2 Demographic variations

We evaluated differences in certifiable-robustness rates across demographic groups in all three datasets. We present results from COMPAS and Adult Income in Fig. 2 (results for Drug Consumption are in the Appendix; they are less interesting due to a lack of representation in the dataset).

**COMPAS.** Fig. 2a shows that under FLIP, White women are robust at a higher rate than any other demographic group, and that Black men and women are the least robust. Notably, for $\text{FLIP}_{0.4\%}$, we are able to certify robustness for 50.4% of White women, but 0% of Black people. There is also a significant gap between White women and White men at this threshold (50.4% vs. 38.8%). We can explain the gaps in certification rates of different subgroups by looking at the training data. In the COMPAS dataset, the same predicate provides the optimal split for every race-gender subgroup, but for White women the resulting split has cost $= 0.41$ versus cost $= 0.46$ for Black people. It is not clear whether this difference stems from sampling techniques or inherent differences in the population, but regardless, the end result is that **predictions made about Black people from decision trees trained on COMPAS are more likely to be vulnerable to data bias**.

To validate that the disparities in certifiable-robustness rates are due to inherent dataset properties rather than the abstraction, we performed random testing by perturbing the COMPAS dataset to try to find robustness counterexamples, i.e., datasets in the bias set that yield conflicting predictions on a given input. We found more counterexamples to robustness for Black people than for White people, which is further evidence for our claim that the robustness disparities are inherent to the dataset.

**Targeted bias models (COMPAS).** If we choose $g \triangleq (race = \text{Black} \wedge label = \text{positive})$ (Fig. 2b) in $\text{FLIP}^g$ to model the real-world situation where structural or individual racism can lead to increased policing and convictions among Black people in the U.S., then there are generally higher robustness rates at moderate bias levels (e.g., ~50% robustness for all demographic groups at 0.4% poisoning). However, as the amount of bias increases, a gap between White and Black certification rates emerges (in exact terms, 32.9% of White test samples are certifiably robust versus 0% of Black test samples starting at 1.1% bias and continuing through, at least, 10.8% bias). It is unclear whether this trend stems from inherent dataset properties, or is due to the over-approximate nature of the abstraction.

By contrast, using $g \triangleq (race = \text{White} \wedge label = \text{negative})$ (Fig. 2c) to describe that White people may be under-policed or under-convicted due to White privilege nearly eliminates discrepancies between demographic groups. In particular, Black men (previously the least-robust subgroup) are the most robust of any population. $\text{FLIP}^g$ and $\text{FLIP}^{g'}$ differ only on how they describe societal inequities:

are White people under-policed, or are Black people over-policed? However, the vast differences in demographic-level robustness rates between FLIP$^g$ and FLIP$^{g'}$ shows that **the choice of predicate is crucial when using targeted bias models**. More experimentation is needed to understand why these results occur, and how consistent they are across different train/test splits of the data. However, our preliminary results indicate that Antidote-P could be a useful tool for social scientists to understand how data bias can affect the reliability of machine-learning outcomes.

**Adult Income.** Fig. 2d shows robustness by demographic group for FLIP. We see that Black men have about a 5% lower robustness rate than other demographic groups and that at higher bias levels, White women also have about a 5% lower robustness rate than White men or Black women. Using FLIP$^g$ where $g = (race = \text{Female} \wedge Label = \text{negative})$ led to similar results (see Appendix).

## 6 Conclusions and broader impacts

We saw that our decision-tree-learner abstraction is able to verify pointwise robustness over large and even infinite bias sets. These guarantees permit increased confidence in the trees' outputs because they certify that data bias has not affected the outcome (within a certain threshold). However, a couple of tricky aspects—and ones that we do not attempt to address—are knowing whether the assumptions underlying the bias model are correct, or whether our bias framework is even capable of representing all instances of real-world bias. If the user does not specify the bias model faithfully, then any proofs may not be representative. Also, our tool only certifies robustness, not accuracy. Therefore, it may certify that a model will always output the *wrong* label on a given data point. This behavior is linked to a shortcoming of many machine-learning audits: our tool cannot determine what is an appropriate use of machine learning. Machine learning is often used to promote and legitimize uses of technology that are harmful or unethical. In particular, we want to call out our use of the COMPAS dataset: we feel that it is illustrative to show how certifiable-robustness rates can vary widely between different demographic groups and be sensitive to subtle shifts in the bias model. However, this use should not be taken as an endorsement for the deployment of recidivism-prediction models.

Another limitation is that our framework can only certify decision-tree learners. In practice, many machine learning applications use more sophisticated algorithms that we do not address here. Future work to generalize our ideas to other machine learning architectures would increase the utility of this style of robustness certification.

Returning to our work, Antidote-P has a place in data scientists' tool-kits as a powerful technique to understand robustness, and potential vulnerabilities, of data bias in decision-tree algorithms. An important direction for future work is to develop effective techniques for falsification of robustness (i.e., techniques to find minimal dataset perturbations that break robustness). We performed initial experiments in this area using brute-force techniques (i.e., randomly perturb data points, train a new decision tree and see whether the test sample's classification changes under the new tree—see the Appendix for more details). The results were promising in that we were able to find counter-examples to robustness for some data points, but there remain many data points that are neither certifiably robust via Antidote-P nor falsified as robust using random testing. Random testing was an interesting proof of concept, but we recommend that the future focus be on developing techniques to identify these dataset perturbations in a more scalable and intelligent way. Other future work could also improve our approach's utility through tightening the analytical bounds, such as by abstracting over a more complex domain than intervals.

## Acknowledgments and Disclosure of Funding

We thank the anonymous reviewers for commenting on earlier drafts and Sam Drews for the generous use of his code. This work is supported by the National Science Foundation grants CCF-1420866, CCF-1704117, CCF-1750965, CCF-1763871, CCF-1918211, CCF-1652140, a Microsoft Faculty Fellowship, and gifts and awards from Facebook and Amazon.

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
