# A    Additional details and definitions

Throughout the appendices, we use square brackets, rather than braces, to denote composite bias models: this is to emphasize that the transformers are ordered, and that alternate orderings often result in distinct bias sets.

**Filtering composite bias models.** Filtering a composite bias model requires us to apply filter piece-wise, i.e., $[\text{MISS}_m^{g_1}, \text{FLIP}_l^{g_2}, \text{FAKE}_k^{g_3}](T)_\phi = [\text{MISS}_m^{g_1 \wedge \phi}, \text{FLIP}_l^{g_2}, \text{FAKE}_k^{g_3}](T_\phi)$.

$\text{pr}^a$ **for FAKE.** Given $c_i$ samples in $T$ with label $i$, we use $k_i = \min(k, c_i)$ and then define

$$\text{pr}_i^a(\text{FAKE}_k(T)) = \left[ \frac{c_i - k_i}{|T| - k_i}, \frac{c_i}{|T| - \sum_{j \neq i} k_j} \right] \tag{11}$$

For the edge case where $c_i = |T|$ and $c_i \leqslant k$ for any $i$, we define $\text{pr}_j^a(T) = [0, 1]$ for all $j \in [1, n]$. A similar edge case applies, when necessary, to the composite definition.

**Optimizing $\text{pr}^a$ when $g$ looks at the label.** If $g$ conditions on the label, then we can improve the precision of $\text{pr}^a$ by defining each component individually. Suppose $g(x, y) = y \in S \wedge g'(x)$, where $S \subset \{1, \cdots, n\}$ and $g'$ is a predicate that only conditions on features.

For $\text{MISS}_m^g$, we define

$$\text{pr}_i^a(\text{MISS}_m^g) = \begin{cases} \left[ \frac{c_i}{|T|}, \frac{c_i + m}{|T| + m} \right] & \text{if } i \in S \text{ and } |S| = 1 \\ \left[ \frac{c_i}{|T| + m}, \frac{c_i + m}{|T| + m} \right] & \text{if } i \in S \text{ and } |S| \geq 2 \\ \left[ \frac{c_i}{|T| + m}, \frac{c_i}{|T|} \right] & \text{else} \end{cases} \tag{12}$$

For $\text{FLIP}_l^g$, we use $l_{a_i} = \min(l, |\{(x, y) \in T \mid y = i \wedge g(x, y)\}|)$ and $l_{b_i} = \min(l, |\{(x, y) \in T \mid y \neq i \wedge g(x, y)\}|)$. Then, we define

$$\text{pr}_i^a(\text{FLIP}_l^g) = \begin{cases} \left[ \frac{c_i - l_{a_i}}{|T|}, \frac{c_i}{|T|} \right] & \text{if } i \in S \text{ and } |S| = 1 \\ \left[ \frac{c_i - l_{a_i}}{|T|}, \frac{c_i + l_{b_i}}{|T|} \right] & \text{if } i \in S \text{ and } |S| \geq 2 \\ \left[ \frac{c_i}{|T|}, \frac{c_i + l_{b_i}}{|T|} \right] & \text{else} \end{cases} \tag{13}$$

For $\text{FAKE}_k^g$, we use $k_{a_i} = \min(k, |\{(x, y) \in T \mid y = i \wedge g(x, y)\}|)$ and $k_{b_i} = \min(k, |\{(x, y) \in T \mid y \neq i \wedge g(x, y)\}|)$. Then, we define

$$\text{pr}_i^a(\text{FAKE}_k^g) = \begin{cases} \left[ \frac{c_i - k_{a_i}}{|T| - k_{a_i}}, \frac{c_i}{|T|} \right] & \text{if } i \in S \text{ and } |S| = 1 \\ \left[ \frac{c_i - k_{a_i}}{|T| - k}, \frac{c_i}{|T| - k_{b_i}} \right] & \text{if } i \in S \text{ and } |S| \geq 2 \\ \left[ \frac{c_i}{|T|}, \frac{c_i}{|T| - k_{b_i}} \right] & \text{else} \end{cases} \tag{14}$$

We prove that the above definitions are sound and precise in Appendix D. If desired, the above definitions can be pieced together to provide a more precise definition for composite bias models. However, we limit ourselves to just the singleton transformers because notation becomes very messy, as we have to keep track of many variables indicating how many data elements satisfy the various conditions.

$\text{pr}^a$ **for composite bias models with multiple versions of the same transformer.** If a bias model contains multiple instances of the same transformer, e.g., $B = [\text{FLIP}_{l_1}^{g_1}, \text{FLIP}_{l_2}^{g_2}]$, we can combine everything into a single transformer. Formally, given

$$B = [\text{MISS}_{m_1}^{g_1'}, \ldots, \text{MISS}_{m_j}^{g_j'}, \text{FLIP}_{l_1}^{g_{j+1}'}, \ldots, \text{FLIP}_{l_p}^{g_{j+p}'}, \text{FAKE}_{k_1}^{g_{j+p+1}'}, \ldots, \text{FAKE}_{k_q}^{g_{j+p+q}'}] \tag{15}$$

we define

$$m = m_1 + \ldots + m_j$$

$$g_1 = g_1' \vee \cdots \vee g_j'$$
$$l_i = \min(l_1 + \ldots + l_p, |\cup_{i \in [1,p]} T_{g_{j+i}(x,y) \wedge y=i}|)$$
$$g_2 = g_{j+1}' \vee \cdots \vee g_{j+p}'$$
$$k_i = \min(k_1 + \ldots + k_p, |\cup_{i \in [1,q]} T_{g_{j+p+i}(x,y) \wedge y=i}|)$$

and

$$g_3 = g_{j+p+1}' \vee \cdots \vee g_{j+p+1}'$$

Then, we can use the formula shown in Equation 8 to compute $\mathsf{pr}^a$. We show in Appendix C that these definitions are sound.

**Size** (size). We define $\mathsf{size}(\mathrm{MISS}_m^g) = [|T|, |T| + m]$, $\mathsf{size}(\mathrm{FLIP}_l^g) = [|T|, |T|]$, and $\mathsf{size}(\mathrm{FAKE}_k^g) = [|T| - k, |T|]$. Putting this all together, we have $\mathsf{size}([\mathrm{MISS}_m^{g_1}, \mathrm{FLIP}_l^{g_2}, \mathrm{FAKE}_k^{g_3}]) = [|T| - k, |T| + m]$.

# B   Proof of optimal composition of transformers

As stated in § 3, when composing transformers we want to apply them in an order that results in the largest composite bias model. To illustrate the concept of composite bias models' relative size, consider $B = [\mathrm{MISS}_1^{g_1}, \mathrm{FLIP}_1^{g_2}]$ and $B' = [\mathrm{FLIP}_1^{g_2}, \mathrm{MISS}_1^{g_1}]$ where $g_1 \triangleq (gender{=}\text{female} \wedge label{=}1)$ and $g_2 \triangleq (gender{=}\text{female})$. I.e., $B$ adds one data point subject to $g_1$ and then flips the label of one data point subject to $g_2$, whereas $B'$ performs these two operations in the opposite order. Under $B$, we can use MISS to add the data point $(x,y) = \langle gender{=}\text{female}, label{=}1 \rangle$ and then use FLIP to change $y$ to 0. However, under $B'$, we cannot alter the point that MISS adds, so $B$ and $B'$ are not equivalent. In this case, $B$ can construct every dataset that $B'$ can construct (but not vice-versa), so we write $B' \subset B$ and say that $B$ is larger than $B'$.

First we consider the case when there are multiple transformers of the same type.

**Theorem 3.** *The bias models $B_1 = [\mathrm{MISS}_{m_1}^{g_1}, \mathrm{MISS}_{m_2}^{g_2}]$ and $B_2 = [\mathrm{MISS}_{m_2}^{g_2}, \mathrm{MISS}_{m_1}^{g_1}]$ are equivalent (and likewise for FAKE and FLIP, as long as no FLIP predicate conditions on the label).*

*Proof.* **Missing data.** The choice of what missing data to add has no bearing on what is already in (or not in) the dataset. Thus we can add $m_1$ elements that satisfy $g_1$ followed by $m_2$ elements that satisfy $g_2$, or do the operators in the reverse order, but the end result is the same.

**Label-flipping.** Suppose $B = [\mathrm{FLIP}_{l_1}^{g_1}, \mathrm{FLIP}_{l_2}^{g_2}]$, where $g_1$ and $g_2$ do not condition on the label. We want to show that $B$ is equivalent to $B' = [\mathrm{FLIP}_{l_2}^{g_2}, \mathrm{FLIP}_{l_1}^{g_1}]$.

Consider an arbitrary $T' \in B(T)$. Each data point in $T'$ is either (1) untouched by $\mathrm{FLIP}_{l_1}^{g_1}$ and $\mathrm{FLIP}_{l_2}^{g_2}$, (2) flipped only by $\mathrm{FLIP}_{l_1}^{g_1}$, (3) flipped only by $\mathrm{FLIP}_{l_2}^{g_2}$, or (4) flipped by both $\mathrm{FLIP}_{l_1}^{g_1}$ and $\mathrm{FLIP}_{l_2}^{g_2}$. If (1), clearly this is obtainable by $B'$ since we did nothing. If (2), then since the data point is untouched by $\mathrm{FLIP}_{l_2}^{g_2}$, the data point can be flipped uninterrupted by $\mathrm{FLIP}_{l_1}^{g_1}$ (similarly for (3)). If (4), then – since neither $g_1$ nor $g_2$ conditions on the label nor specifies what the new label can be – we can still flip the label twice and end up with the same configuration. The same arguments hold had we started with $T'' \in B'$. Therefore, $B$ and $B'$ are equivalent.

**Fake data.** The argument for fake data is similar. $\qquad\square$

We can extend the proof of Theorem 3 to arbitrarily many transformers of the same type.

Note that if FLIP conditions on the label, this proof does not hold. To continue with the terminology from the proof, if $g_i \triangleq (label{=}a)$, then applying $\mathrm{FLIP}_{l_j}^{g_j}$ first to some element $(x, a)$ yields $(x, a')$, which may no longer eligible to be flipped by $\mathrm{FLIP}_{l_i}^{g_i}$.

Next, we show that there is an optimal way to compose transformers of different types. We define *optimal* as largest, that is, some $B'$ is optimal compared to $B$ if $B \subseteq B'$. In other words, this notation says that every dataset created by $B$ can also be created by $B'$. For the next theorem and its proof we assume there is only one instance of each transformer type; however, in conjunction with Theorem 3 we can extend it to include multiple instances of the same transformer type.

**Theorem 4.** $B = [\mathrm{MISS}, \mathrm{FLIP}, \mathrm{FAKE}]$ *is the optimal order to apply the transformers* MISS, FLIP, *and* FAKE *(i.e., any other ordering $B'$ of these transformers will satisfy $B' \subseteq B$).*

*Proof.* We will show that other orderings of MISS, FLIP, FAKE do not produce any biased datasets that do not also occur in [MISS, FLIP, FAKE]. For conciseness, we will write [MISS, FLIP, FAKE] as OPT.

1. [MISS, FAKE, FLIP]: We consider the set of datasets $S$ achieved after applying MISS, FAKE, and then FLIP. Fix an arbitrary $T' \in S$. $T'$ was constructed from $T$ by some sequence of adding, removing, and flipping data points. We have these categories for (potential) data points in $T'$: (1) untouched data points, (2) added data points, (3) added then removed data points, (4) added then flipped data points, (5) removed data points, and (6) flipped data points. (1), (2), (5), and (6) apply single (or no) operators, so clearly are also attainable through OPT. MISS occurs before both FLIP and FAKE in OPT, so (3) and (4) are attainable, as well.

2. [FLIP, MISS, FAKE]: We consider the set of datasets $S$ achieved after applying FLIP, MISS, and then FAKE. Fix an arbitrary $T' \in S$. $T'$ was constructed from $T$ by some sequence of flipping, adding, and removing data points. We have these categories: (1) untouched data points, (2) flipped data points, (3) flipped then removed data points, (4) added data points, (5) added then removed data points, and (6) removed data points. (1), (2), (4), and (6) apply single (or no) operators, so clearly they are also attainable through OPT. Since flipping and adding each come before removing in OPT, (3) and (5) are obtainable as well.

3. [FLIP, FAKE, MISS]: We consider the set of datasets $S$ achieved after applying FLIP, FAKE, and then MISS. Fix an arbitrary $T' \in S$. $T'$ was constructed from $T$ by some sequence of flipping, removing, and adding data points. We have these categories: (1) untouched data points, (2) flipped data points, (3) flipped then removed data points, (4) removed data points, and (5) added data points. (1), (2), (4), and (5) apply single (or no) operators, so clearly they are also attainable through OPT. Since flipping comes before removing in OPT, (3) is obtainable as well.

4. [FAKE, MISS, FLIP]: We consider the set of datasets $S$ achieved after applying FAKE, MISS, and then FLIP. Fix an arbitrary $T' \in S$. $T'$ was constructed from $T$ by some sequence of removing, adding, and flipping data points. We have these categories: (1) untouched data points, (2) removed data points, (3) added data points, (4) added then flipped data points, and (5) added data points. (1), (2), (4), and (5) apply single (or no) operators, so clearly they are also attainable through OPT. Since flipping comes before removing in OPT, (3) is obtainable as well.

5. [FAKE, FLIP, MISS]: We consider the set of datasets $S$ achieved after applying FAKE, FLIP, and then MISS. Fix an arbitrary $T' \in S$. $T'$ was constructed from $T$ by some sequence of removing, flipping, and adding data points. We have these categories: (1) untouched data points, (2) removed data points, (3) flipped data points, and (4) added data points, Each of these apply single (or no) operators, so clearly they are also attainable through OPT.

We were not able to construct a dataset not also in OPT through any other ordering of the operators, therefore, OPT is optimal. □

## C  Proofs of soundness

**Proof of Theorem 4.2.** $\mathrm{pr}^a$ is sound.

*Proof.* We show MISS as a simple example to illustrate our approach, and then we show the proof for composite bias. We omit the proofs for FLIP and FAKE because they (like MISS) are special cases of composite.

**Missing data.** Given a dataset $T$ with $n$ classes, suppose our bias set is $\mathrm{MISS}_m^g(T)$. Furthermore, suppose that $c_i$ samples in $T$ have label $i$. We define $m_i \in [0, m]$ to be the number elements we add with label $i$, and $m_i' = \sum_{j \neq i} m_j$. Then, we can write the proportion of $i$'s as a function

$$F(m_i, m_i') = \frac{c_i + m_i}{|T| + m_i + m_i'} \tag{16}$$

The partial derivatives of $F$ have values $\frac{\delta F}{\delta m_i} > 0$ and $\frac{\delta F}{\delta m_i'} < 0$ over the entire domain $[0, m]$, therefore, any conclusions we draw over $\mathbb{R}$ will also apply over the discrete integer domain. Therefore, to minimize $F$ we choose $m_i = 0$ and $m_i' = m$, and do the reverse to maximize $F$. Thus, $F_{\min} = \frac{c_i}{|T|+m}$ and $F_{\max} = \frac{c_i+m}{|T|+m}$. Since $[F_{\min}, F_{\max}] \subseteq \mathrm{pr}_i^a(\mathrm{MISS}_m^g(T))$, $\mathrm{pr}^a$ is sound.

**Composite.** Given a dataset $T$ with $n$ classes, suppose that our bias model is $B = [\text{MISS}_m^{g_1}, \text{FLIP}_l^{g_2}, \text{FAKE}_k^{g_3}]$. Furthermore, suppose that in $T$, $c_i$ samples have label $i$.

First, we consider how many elements we can add, flip, or remove of each label. Under MISS, we can add $m$ of label $i$ for all $i$. Under FLIP, we can flip up to $l$ labels from label $i$ to some $j \neq i$, assuming that there are at least $l$ elements $(x, y)$ that satisfy $g_2(x, y)$ and $y = i$. Similarly, the maximum number of labels we can flip from any label $j \neq i$ to $i$ is bounded both by $l$ and by the number of elements that satisfy $g_2$ and have label $j$. Note that the elements we can flip are not just limited to $T$: we can also flip any of the newly-added $m$ elements. Formally, we define $l_{a_i} = \min(|\{(x, y) \in T \mid y = i \wedge g_2(x, y)\}| + m, l)$, and $l_{b_i} = \min(|\{(x, y) \in T \mid y \neq i \wedge g_2(x, y)\}| + m, l)$. Similarly, under FAKE we must also consider the elements added by MISS and those flipped from $j \neq i$ to $i$ by FLIP. Therefore, we define $k_{a_i} = \min(|\{(x, y) \in T \mid y = i \wedge g_3(x, y)\}| + m + l_{b_i}, k)$ and $k_{b_i} = \min(|\{(x, y) \in T \mid y \neq i \wedge g_3(x, y)\}| + m + l_{a_i}, k)$.

We will show that under the specified bias model, the proportion of $i$'s in the dataset is always between $(c_i - l_{a_i} - k_{a_i})/(|T| - k_{a_i} + m)$ and $(c_i + m + l_{b_i})/(|T| - k_{b_i} + m)$.

Intuitively, we consider how to modify the proportion of $i$'s in $T$. This proportion decreases by (1) flipping elements from class $i$ to some class $j \neq i$, (2) removing elements of class $i$, and (3) adding elements of a class other than $i$. Therefore, the fraction is minimized by doing (1), (2), and (3) as much as the bias model allows.

Formally, let $m_i \in [0, m]$ be the number of elements added with label $i$, $m_i' \in [0, m] = \sum_{j \neq i} m_j$, $l_i \in [0, l_{a_i}]$ be the number of elements flipped from class $i$ to any other class, $l_i' \in [0, l_{b_i}] = \sum_{j \neq i} l_i$, $k_i \in [0, k_{a_i}]$ be the number of elements removed with label $i$, $k_i' \in [0, k_{b_i}] = \sum_{j \neq i} k_i$.

Then, we can write the proportion of $i$'s as a function

$$F(m_i, m_i', l_i, l_i', k_i, k_i') = \frac{c_i + m_i - l_i + l_i' - k_i}{|T| + m_i + m_i' - k_i - k_i'} \tag{17}$$

We consider the partial derivative of $F$ with respect to each variable. For all input in the domain, we have $\frac{\delta F}{\delta m_i} > 0$, $\frac{\delta F}{\delta m_i'} < 0$, $\frac{\delta F}{\delta l_i} < 0$, $\frac{\delta F}{\delta l_i'} > 0$, $\frac{\delta F}{\delta k_i} < 0$, and $\frac{\delta F}{\delta k_i'} > 0$.

Note that each partial derivatives is monotone over all values in the domain. Thus, they are also monotone over integers, so any conclusions we yield over the real numbers can be relaxed to integers, as well. To minimize $F$, we will maximize each variable whose partial derivative is negative, and minimize each variable whose partial derivative is positive. That is, we choose $m_i = 0$, $m_i' = m$, $l_i = l_{a_i}$, $l_i' = 0$, $k_i = k_{a_i}$, and $k_i' = 0$ to minimize $F$, yielding

$$F_{\min} = \frac{c_i - l_{a_i} - k_{a_i}}{|T| + m - k_{a_i}}$$

Conversely, to maximize $F$ we maximize each variable whose partial derivative is positive and minimize each variable whose partial derivative is negative, yielding

$$F_{\max} = \frac{c_i + m + l_{b_i}}{|T| + m - k_{b_i}}$$

$[F_{\min}, F_{\max}] \subseteq \text{pr}_i^a(B(T))$, therefore, $\text{pr}^a$ is sound.

**Multiple composite bias models.** Suppose

$$B = [\text{MISS}_{m_1}^{g_1'}, \ldots, \text{MISS}_{m_j}^{g_j'}, \text{FLIP}_{l_1}^{g_{j+1}'}, \ldots, \text{FLIP}_{l_p}^{g_{j+p}'}, \text{FAKE}_{k_1}^{g_{j+p+1}'}, \ldots, \text{FAKE}_{k_q}^{g_{j+p+q}'}]$$

and

$$B' = \left[\text{MISS}_{\sum_{i \in [1,j]} m_i}^{\bigvee_{i \in [1,j]} g_i}, \text{FLIP}_{\sum_{i \in [1,p]} l_i}^{\bigvee_{i \in [j+1, j+p]} g_i}, \text{FAKE}_{\sum_{i \in [1,q]} k_i}^{\bigvee_{i \in [j+p+1, j+p+q]} g_i}\right]$$

We want to show that if $(x, y) \in T \cup T'$ is altered by $B$, it can be altered by $B'$ (in other words, we want to show that $B \subseteq B'$).

Case 1: $(x, y)$ was added by a transformer $\text{MISS}_{m_i}^{g_i}$ for $i \in [1, j]$. Therefore, $g_i(x)$ and $(\bigvee_{i \in [1,j]} g_i)(x)$, as well, so $x$ can be added by $\text{MISS}_{\sum_{i \in [1,j]} m_i}^{\bigvee_{i \in [1,j]} g_i}$.

Case 2: $(x, y) \in T$ was flipped to $(x, y') \in T'$ by $\text{FLIP}_{l_i}^{g_{j+i}}$ for $i \in [1, p]$. This means that $g_{j+i}(x)$, so by extension, $(\bigvee_{i \in [j+1, j+p]} g_i)(x)$, which means that $x$'s label can be flipped by $\text{FLIP}_{\sum_{i \in [1,p]} l_i}^{\bigvee_{i \in [j+1, j+p]} g_i}$.

Case 3: $(x, y) \in T$ was removed by $\text{FLIP}_{k_i}^{g_{j+p+i}}$ for $i \in [1, q]$. This means that $g_{j+p+i}(x)$, and thus $\bigvee_{i \in [j+p+1, j+p+q]} g_i(x)$. Therefore $(x, y)$ can be removed by $\text{FAKE}_{\sum_{i \in [1,q]} k_i}^{\bigvee_{i \in [j+p+1, j+p+q]} g_i}$.

To conclude, any modification to $T$ that we can make under $B$ is also attainable under $B'$, therefore, if $B'$ is sound, then $B$ is sound, as well. $\qquad\square$

**Proposition 1.** *Abstract filtering is sound.*

*Proof.* To prove soundness for filtering, we need to show that $T' \in B(T) \implies T'_\phi \in B(T)_\phi$.

**Missing data.** Consider $T' \in \text{MISS}_m^g(T)$. Since $T' \in \text{MISS}_m^g$, we have $T' = T \cup S$ where $|S| \leqslant m$ and $\forall (x, y) \in S. g(x, y)$. Therefore, $T'_\phi = T_\phi \cup S_\phi$. Since $S_\phi \subseteq S$, then $|S_\phi| \leqslant m$ and $\forall (x, y) \in S_\phi. g(x, y)$. then $T_\phi \cup S_\phi \in \text{MISS}_m^{g \wedge \phi}$, satisfying the claim.

**Label flipping.** Consider $T' \in \text{FLIP}_l^g(T)$. Since $T' \in \text{FLIP}_l^g(T)$, we know that $T' = R \cup S$ where $R \subseteq T$, $|S| \leqslant l$, and $T = R \cup \{(x, y) \mid (x, y') \in S\}$. Additionally, we have $(x, y) \in S \implies g(x, y)$. Consider $T'_\phi = R_\phi \cup S_\phi$. Since $R \subseteq T$, then $R_\phi \subseteq T_\phi$. Since $(x, y) \in S \implies (x, y') \in T$ and $\phi$ does not condition on the label, then $(x, y) \in S_\phi \implies (x, y') \in T_\phi$. Since $S_\phi \subseteq S$, we have $|S_\phi| \leqslant l$. In total, this means that $T_\phi \in \text{FLIP}_l^g(T_\phi)$.

**Fake data.** See [16].

**Composite.** Suppose $B = [\text{MISS}_m^{g_1}, \text{FLIP}_l^{g_2}, \text{FAKE}_k^{g_3}]$. We want to show that $B(T)_\phi \subseteq B'(T_\phi)$, where $B' = [\text{MISS}_m^{g_1 \wedge \phi}, \text{FLIP}_l^{g_2}, \text{FAKE}_k^{g_3}]$. Consider $T' \in B(T)$. Each $(x, y) \in T'$ satisfies either (i) $(x, y) \in T$, (ii) $(x, y') \in T$ (for $y' \neq y$), or (iii) $(x, y') \notin T$. If (i), then $x \in T'_\phi \implies x \in T_\phi$, and likewise for $\neg \phi$. If (ii), $x \in T'_\phi \implies x \in T_\phi$ (since $\phi$ ignores the label), and likewise for $\neg \phi$. If (iii), the $x$ was added by MISS. $x \in T'_\phi \implies \phi(x) \implies x$ can be added by $\text{MISS}_m^{g_1 \wedge \phi}$, and if $x \notin T_\phi$, this means that $x$ cannot be added by $\text{MISS}_m^{g_1 \wedge \phi}$. Finally, there is a fourth category of elements: those in $T \setminus T'$. If $x \in T \setminus T'$, then $x$ was removed by FAKE. If $\phi(x)$, then $x$ can be removed from $T$ to make $T'$, otherwise, $x$ cannot be removed from $T$, so it must also be contained in $T'$.

Thus we have shown that $T'_\phi$ can be constructed from $T_\phi$ using $B'$, therefore, filtering is sound. $\quad\square$

**Proposition 2.** $\text{imp}^a$ *is sound*

*Proof.* To show that $\text{imp}^a$ is sound, we must show that $T' \in B(T) \implies \text{imp}(T') \in \text{imp}^a(B(T))$. By § 4.2, $\text{pr}(T') \in \text{pr}(B(T))$. It follows from interval arithmetic $\text{imp}(T') \in \text{imp}^a(B(T))$. $\quad\square$

**Proposition 3.** $\text{size}$ *is sound.*

*Proof.* Given a bias model $B = [\text{MISS}_m^{g_1}, \text{FLIP}_l^{g_2}, \text{FAKE}_k^{g_3}]$ and a dataset $T$, we can write the size of $T' \in B(T)$ as $|T| - k' + m'$, where $k' \in [0, k]$ and $m' \in [0, m]$. Clearly, $|T| - k' + m'$ is minimized by choosing $k' = k$ and $m' = 0$, and maximized by choosing $k' = 0$ and $m' = m$. Since $|T| - k \in \text{size}(B)$ and $|T| + m \in \text{size}(B)$, we see that size is sound. $\quad\square$

**Proposition 4.** $\text{cost}^a$ *is sound.*

*Proof.* We need to show that if $T' \in B(T)$ and $\phi \in \Phi$, then $\text{cost}(T', \phi) \in \text{cost}^a(B(T), \phi)$. By Proposition 1, we know that $T'_\phi \in B(T)_\phi$, which means that by Proposition 3, $|T'_\phi| \in \text{size}(B(T)_\phi)$, and similarly we can derive that $|T'_{\neg \phi}| \in \text{size}(B(T)_{\neg \phi})$. Additionally, by Proposition 2, we know that $\text{imp}(T'_\phi) \in \text{imp}^a(B(T)_\phi)$ and $\text{imp}(T'_{\neg \phi}) \in \text{imp}^a(B(T)_{\neg \phi})$. By the rules of interval arithmetic, if $a \in [a_0, a_1]$ and $b \in [b_0, b_1]$, then $ab \in [a_0, a_1] \times [b_0, b_1]$ and $a + b \in [a_0, a_1] + [b_0, b_1]$. Therefore we can conclude that $\text{cost}(T', \phi) \in \text{cost}^a(B(T), \phi)$, i.e., $\text{cost}^a$ is sound. $\quad\square$

**Proposition 5.** $\text{split}^a$ *is sound.*

*Proof.* We want to show that if $T' \in B(T)$, then $\mathsf{split}(T') \in \mathsf{split}^a(B(T))$.

Suppose $\gamma = \mathsf{split}(T')$. Then, $\forall \phi \in \Phi$, $\mathsf{cost}(T', \phi) \geq \mathsf{cost}(T', \gamma)$.

Define $\phi^*$ such that $lub = ub(\mathsf{cost}(T, \phi^*))$, where $ub$ takes the upper bound of an interval. Since $\phi^* \in \Phi$, $\mathsf{cost}(T', \gamma) \leqslant \mathsf{cost}(T', \phi^*)$. By Proposition 4, cost is sound, therefore $\mathsf{cost}(T', \gamma) \in \mathsf{cost}^a(B(T), \gamma)$ and $\mathsf{cost}(T', \phi^*) \in \mathsf{cost}^a(B(T), \phi^*)$. And thus we have $lb(\mathsf{cost}^a(B(T), \gamma)) \leqslant ub(\mathsf{cost}^a(B(T), \phi^*)) = lub$. Thus, $\gamma \in \mathsf{split}^a(B(T))$. $\qquad\square$

**Proof of Theorem 2.**

*Proof.* Given $\Phi^a = \mathsf{split}^a(B(T))$, if $|\mathsf{infer}^a(B(T), \Phi^a, x)| = 1$, then we know

$$\exists y. \forall \phi \in \Phi^a. \begin{cases} \text{if } \phi(x) \text{ then } \mathrm{argmax}_i \, \mathsf{pr}_i^a(B(T)_\phi) = y \\ \text{if } \neg\phi(x) \text{ then } \mathrm{argmax}_i \, \mathsf{pr}_i^a(B(T)_{\neg\phi}) = y \end{cases}$$

Given $T' \in B(T)$, we know from Proposition 5 that $\mathsf{split}(T') \in \mathsf{split}^a(T) = \Phi^a(T)$. Therefore, $\mathsf{infer}(T', \mathsf{split}(T'), x) = y$, so the algorithm is robust on $x$. (Note that we defined $B(T)$ such that $T \in B(T)$, therefore, the original prediction is also $y$.) $\qquad\square$

# D  Precision

Intuitively, an abstraction is precise if the abstraction cannot be improved. Formally, our abstraction is precise iff, it is sound and given $\mathsf{pr}_i^a(B(T)) = [a_i, b_i]$, then for each $i$ there is some $T' \in B(T)$ such that $\mathsf{pr}_i(T') = a_i$ and some $T'' \in B(T)$ such that $\mathsf{pr}_i(T'') = b_i$.

**Theorem 5.** $\mathsf{pr}^a$ *is precise for missing data, label-flipping, and fake data.*

*Proof.* **Missing data (non-targeted).** In the proof of Theorem 4.2, we show that the minimum proportion of $i$'s is $\frac{c_i}{|T|+m}$ and the maximum proportion of $i$'s is $\frac{c_i+m}{|T|+m}$. Since these bounds are equal to $\mathsf{pr}^a$'s minimum and maximum, the interval is precise.

Proofs for label-flipping and fake data (non-targeted) similarly follow from Theorem 4.2.

**Targeted.** Next, we will show that the definitions of $\mathsf{pr}^a$ provided in Equations 12-14 are precise. First, we must show that these definitions are sound (as soundness is a prerequisite for precision).

To show that Eq. (12) is sound, we need to consider three cases (we use $S$ such that $g(x,y) = y \in S \wedge g'(x)$): first, $i \in S$ and $|S| = 1$. In this case, we can add up to $m$ elements of class $i$, and no elements of class $j \neq i$. Therefore, the minimum proportion of $i$'s is the original proportion: $\frac{c_i}{|T|}$, and the maximum is $\frac{c_i+m}{|T|}$. Second, $i \in S$ and $|S| \geq 2$. In this case, we can add elements with label $i$ but we can also add elements with label $j \neq i$. As such, the minimum proportion of $i$'s is achieved by adding $m$ elements with label $j$, and the maximum proportion of $i$'s is achieved by adding $m$ elements with label $i$. Therefore, the minimum proportion of $i$'s is $\frac{c_i}{|T|+m}$ and the maximum proportion of $i$'s is $\frac{c_i+m}{|T|+m}$.

Proofs of soundness for Equations 13 and 14 follow similarly.

To show precision, note that in the soundness proof we described exactly how to achieve the minimum and maximum bounds of $\mathsf{pr}_i^a$. As such, we have shown that the $\mathsf{pr}^a$ definition in Eq. (12) is precise. (Similar conclusions can be drawn based on the proofs for label-flipping and fake data.) $\qquad\square$

$\mathsf{pr}^a$ is not precise for composite bias because the auxiliary variables $l_{a_i}, l_{b_i}, k_{a_i}$, and $k_{b_i}$ are over-approximate. As a motivating example, consider a bias model $[\mathrm{MISS}_1, \mathrm{FLIP}_2^g]$ over dataset $T$ with classes $\{0, 1\}$ where $c_0 = 2$, $c_1 = 5$, and $|\{(x,y) \in T : g(x,y) \wedge i = 0\}| = 1$. By definition, $l_{a_0} = 2$, yielding a minimum proportion of 0's to be $\frac{2-2}{7+2} = 0$. However, the precise lower bound is $\frac{1}{9}$: to minimize the proportion of 0's, we add 2 elements with label 1 and flip the element that satisfies $g$ from label 0 to label 1.

Table 2: Certification rates of MNIST 1/7 binary for various bias models, using decision trees of depth 2. The composite bias models show cumulative bias, e.g., 0.2% MISS + FAKE bias equates to 0.1% bias of each MISS and FAKE. Note that the scale for perturbation set size is slightly different (larger) than that in Table 1.

| Bias type | Dataset | Bias amount as a percentage of training set size | | | | | |
|---|---|---|---|---|---|---|---|
| | | 0.05 | 0.1 | 0.2 | 0.4 | 0.7 | 1.0 |
| MISS | MNIST-1-7 binary | 100.0 | 100.0 | 93.0 | 88.0 | 85.0 | 68.0 |
| FLIP | MNIST-1-7 binary | 100.0 | 95.0 | 88.0 | 70.0 | 63.0 | 38.0 |
| MISS + FAKE | MNIST-1-7 binary | 100.0 | 100.0 | 93.0 | 88.0 | 85.0 | 68.0 |
| MISS + FLIP | MNIST-1-7 binary | 100.0 | 95.0 | 91.0 | 87.0 | 67.0 | 62.0 |
| Perturbation set size: | | $< 10^{10}$ | $< 10^{100}$ | $< 10^{1000}$ | $< 10^{10000}$ | $> 10^{10000}$ | infinite |

Table 3: Time and memory requirements for certifying a single test sample under different FLIP bias models on Adult Income with depth=2.

| | Poisoning Amount (%) | | | | | |
|---|---|---|---|---|---|---|
| | 0.1 | 0.2 | 0.3 | 0.4 | 0.5 | 0.6 |
| Time (s.) | 0.60 | 73.9 | 210 | 810 | 1800 | 5200 |
| Memory (GB) | 0.01 | 0.8 | 3.6 | 9.7 | 21 | 60 |

# E  Additional experimental data

## E.1  MNIST 1/7 Binary

We used MNIST 1/7 (the limitation of MNIST to just 1's and 7's, with training $n$=13,007, as has been used in works including [16, 36]). We round each pixel to 0 or 1 (i.e., used a black-and-white image rather than a grayscale one). The accuracy of MNIST 1/7 binary (97.4% at depth 2) is comparable to that of MNIST 1/7, but the time and memory requirements on Antidote-P are much less.

Table 2 shows effectiveness data for MNIST 1/7 binary. We see that we are able to achieve high robustness certification rates, despite incredibly large perturbation set sizes. Notable, for $MISS_{0.1\%}$, we achieve 100% robustness even with a perturbation set size of over $10^{3058}$, and for $MISS_{1\%}$, we achieve 68% robustness with a perturbation set size larger than $10^{30460}$.

## E.2  Performance

We performed experiments on an HTCondor system, allowing us to perform many experiments in parallel. Each experiment ran robustness tests on a given bias model and dataset for between one and 1000 test samples (depending on the bias model and dataset). We used a single CPU for each experiment, and requested between 1 and 96GB of memory, depending on the bias model and dataset.

**Bias model.** Time and memory requirements increase exponentially as the amount of bias increases, as shown in Table 3 for the Adult Income dataset under the FLIP bias model. Other datasets typically required less than 100 ms per test sample. Additionally, bias models that yield lower certifiable robustness for a given bias threshold have correspondingly larger time and memory requirements (e.g., 810s and 9.7GB of memory to yield 34.8% robustness for $FLIP_{0.4\%}$ as compared with 77s and 1.3GB of memory to yield 60.3% robustness for $MISS_{0.4\%}$).

**Datasets.** The size and complexity of the feature space is most influential in determining time and memory requirements. Experiments on the Adult Income dataset were more resource-intensive than those on Drug Consumption or COMPAS, a fact that can be explained by Adult Income having more unique feature values than the other datasets (22,100 for Adult Income vs. 219 for Drug Consumption and 53 for COMPAS). For each unique value of any feature, the algorithm checks an additional predicate, which explains the additional time and memory needs.

Table 4: Robustness certification rates of COMPAS and Drug-Consumption datasets under FLIP for different decision-tree depths.

| Poisoning amount (%) | COMPAS | | | Drug Consumption | | |
|---|---|---|---|---|---|---|
| | depth 1 | 2 | 3 | depth 1 | 2 | 3 |
| 0.10 | 71.5 | 51.5 | 34.0 | 94.5 | 83.8 | 8.5 |
| 0.20 | 47.8 | 27.7 | 23.9 | 94.5 | 55.9 | 4.5 |
| 0.50 | 9.3 | 2.5 | 0.7 | 85.1 | 27.5 | 0.5 |
| 1.00 | 3.0 | 0.7 | 0 | 7.1 | 0.8 | 0 |

**Complexity of decision-tree algorithm.** Increasing the depth of the decision tree not only requires additional time to essentially re-run the algorithm at each internal node, but also leads to lower certifiable-robustness rates, as shown in Table 4. This is because we must assume worst-case bias in each node. Intuitively, a depth 2 tree with 0.1% bias may initially split the data into two children nodes, each with 50% of the data. Our abstraction captures both the case where all bias occurs in the left child, and the case when all bias occurs in the right child. Therefore, we end up with an effective bias rate of 0.2% in either child, yielding lower robustness.

### E.3  Additional experimental data

**General data.** Fig. 3 shows the certifiable robustness rates for each dataset and each main bias model (MISS, FLIP, MISS + FAKE, and MISS + FLIP).

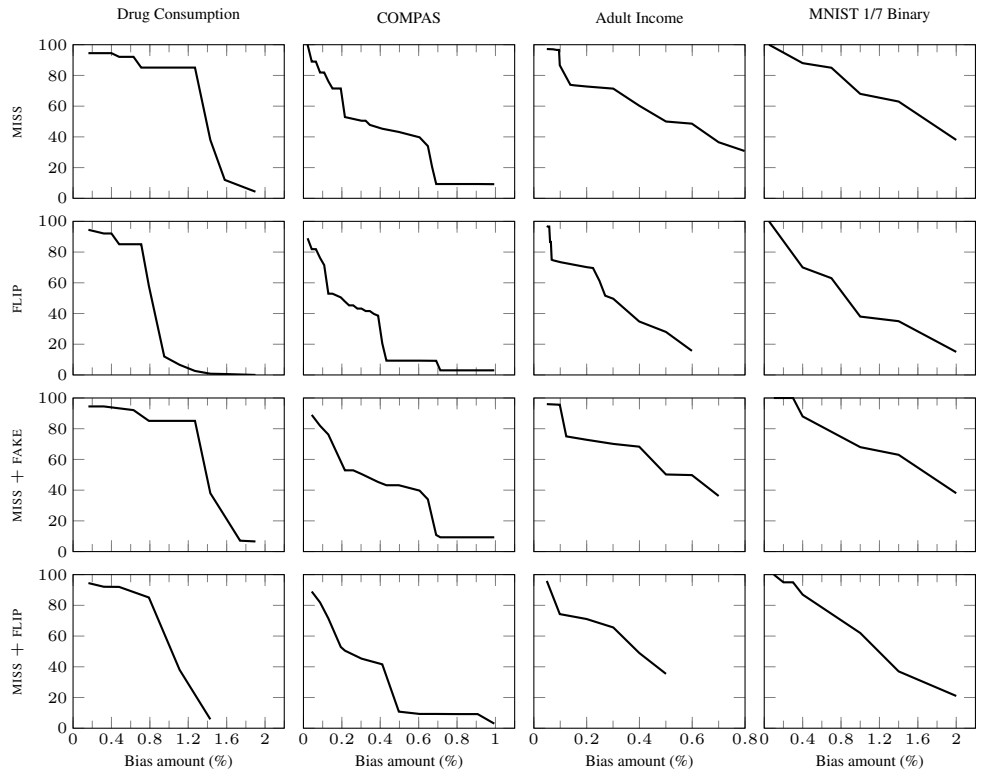

Figure 3: Certifiable robustness (shown on y-axis as a percentage of test data) for various datasets under different bias models.

**Demographic data.**

Fig. 4 shows robustness levels stratified by demographic groups on various bias models. We see that COMPAS under MISS (Fig. 4b) displays similar robustness gaps to FLIP; namely, White people, and particularly White women, are robust at a higher rate than Black people. Adult Income under

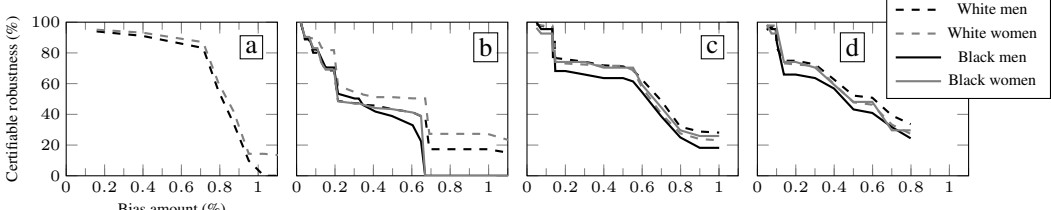

Figure 4: Left to right: Certifiable robustness by demographic group on (a) Drug Consumption under FLIP (NOTE: on this graph, the dark dotted line is all men, and the light dotted line is all women); (b) COMPAS under MISS; (c) Adult Income under FLIP$^g$ where $g \triangleq (gender = $ Female $\wedge$ $label = $ negative); (d) Adult Income under MISS

Table 5: Number of elements with counterexamples to robustness after 10,000 iterations of random testing from a subset of 100 test samples from COMPAS. All bias models are FLIP, and bias level refers to number of affected elements as a percentage of training dataset size.

| Bias level | # of elements with counterexample |
|---|---|
| 0.5 | 4 |
| 1.0 | 15 |
| 2.0 | 27 |
| 3.0 | 32 |

MISS (Fig. 4c) and under FLIP$^g$ where $g \triangleq (gender = $ female $\wedge$ $label = $ negative) (Fig. 4d) behaves similarly to Adult Income under FLIP (Fig. 2). That is, all demographic groups have roughly comparable robustness rates. Drug consumption under FLIP (Fig. 4a) yields comparable robustness rates between men and women (we do not graph robustness rates by racial group because the dataset is over 91% White).

### E.4  Additional details on random testing

On a random subset of 100 test elements from the COMPAS dataset, we tested 10,000 dataset perturbations under FLIP$_{0.5\%}$, FLIP$_{1\%}$, FLIP$_{2\%}$, and FLIP$_{3\%}$. The number of elements for which we found a counter-example to robustness (i.e., a dataset perturbation that resulted in a different classification) is shown in Table 5. We see that we are able to find counterexamples to robustness for a non-trivial portion of test samples. However, the gap between certified-robustness and proved-non-robust rates is still wide (the gap ranges from 86.7% for FLIP$_{0.5\%}$ to 68.0% for FLIP$_{3\%}$). As a result, there are many test samples that we cannot prove robustness for, but cannot find counterexamples for either. Future work to use a more precise abstract domain, or to better identify counterexamples to robustness could help to narrow this gap.

Breaking down the results for FLIP$_{3\%}$ further, we found counterexamples to robustness for 50% of Black women, 37% of Black men, 29% of White women, and 27% of White men. Similarly, we found more counter-examples to robustness for Black people using other bias models. The empirical result of having more counterexamples for test instances representing Black people combined with the fact that we are able to certify a smaller percentage of test instances representing Black people (§ 5.2) suggests that the robustness differences are inherent to the data, rather than a property of the abstraction.