# OpenReview forum: "Certifying Robustness to Programmable Data Bias in Decision Trees"
_NeurIPS.cc/2021/Conference — NeurIPS 2021 Poster_

### Official Review · Reviewer_FQR1 · 2021-07-15

**Rating:** 7
**Confidence:** 4

**Summary:**

The paper considers a method of certifying decision trees for pointwise robustness to data bias. More specifically, the paper introduces a language to encapsulates different types of data bias and then introduces a criterion for robustness for the CART algorithm.

Main Contributions:
 - Sec 3: A language to characterise data bias.
 - Sec 4: The transformed components of the decision tree algorithm.
 - Experimental evaluation of the proposed certification.

**Ethical Concerns:**

No ethical concerns.

**Limitations And Societal Impact:**

The authors have adequately addressed the limitations and potential negative societal impact.

**Main Review:**

Overall, the paper presents an interesting method for certifying robustness for CART decision trees. The paper is well written and the running examples is greatly appreciated. The experimental section also helps elucidate how the certification process can be used to learn more about the datasets. The only concern is the translatability of such a tool, i.e., the simple change in splitting entropy.

Originality & Significance: The paper presents a novel method of certifying CART decision trees. Furthermore, the experiments show how the certification can be used to learn more about datasets. The significance is hurt by how the certification approach can only be used for a specific decision tree setup.

Quality & Clarity: The theoretical and empirical results seem to be sound. The paper is well written with high clarity.

Strengths:
 - The "language" of data bias models is interesting (and could perhaps have use in distributional robustness outside of certification).
 - The certification (Thm 2) and the abstract transformers provide a nice counter part to the regular CART components.
 - The experiments and discussion presents useful inferences for the datasets.

Suggestions/Weaknesses/Questions:
 - Is there a concrete link between bias models and (say, group) fairness metrics? (i.e., is there an example of equating approximate equalized odds with a bias model?)
 - The pointwise robustness condition seems to be quite restrictive. Is there an easy generalization for approximate pointwise robustness?
 - I think it might be worth reiterating that CART decision trees are the target of certification in Sec 6/limitations.

**Time Spent Reviewing:**

7

---

> ### Author Response · Authors · 2021-08-10
> **response to reviewer feedback**
>
> We thank the reviewer for their review! In response to specific questions/comments:
>
> **Connections to fairness metrics** - Robustness to bias is orthogonal to popular fairness metrics like equalized odds and individual fairness. Such metrics are properties of a generated model, often with respect to ground truth data (e.g., in the case of equalized odds). However, our data-bias robustness notion is a property of the learning algorithm. A learning algorithm may produce a fair model, but it may not be robust to data bias.
>
> Exploring deeper connections and interplay between the two notions is indeed an interesting avenue for future work.
>
> **Pointwise robustness** - We are not completely sure what the reviewer means, so we are including responses to two interpretations of the question.
>
> * Interpretation 1: Can we generalize our process to certify whether a subset of the input space is robust? For example, instead of using a sample $x=(1, 0, 1)$ as the certification target, we would use, for example, $x’=([1,1],[0,0],[0,1])$, corresponding to all data points with feature 1$=1$, feature 2$=0$, and feature 3 $\in [0,1]$.
>
>   * Our framework and implementation don’t support this currently, but we don’t see any technical reason why this extension wouldn’t be possible. (When building the decision tree, we would just consider all branches that could be active for any test sample within the certification target).
>
> * Interpretation 2: Can we generalize our algorithm to provide a probabilistic guarantee, rather than an exact one? E.g., can we certify that for 99% of the datasets in the perturbation space, the output of the algorithm will be consistent for a fixed test input?
>
>   * We can envision achieving this through one of two avenues: first, we could impose a probability distribution on the perturbation space and certify that all datasets that occur in the probability distribution with high-enough probability result in a robust prediction. Second, we could randomize the learning algorithm itself and provide a probabilistic guarantee in a similar way to what is achieved by differential privacy. Neither of these approaches seem trivial, but they strike us as interesting ideas for future work.
>
>   * Finally, we also want to emphasize that the exactness of our certification process is one thing that differentiates it from other automated verification tools that use statistical approaches (see paragraph 2 of the Related Work section). But the extensions in the above paragraph would still be novel, because no other work that we are aware of uses abstract interpretation with probabilistic guarantees to verify poisoning robustness.
>
> **Limitations of the CART algorithm** - we acknowledge that CART is not suitable for many tasks, and we agree that this approach will have additional utility if it can be adapted to other ML architectures in the future. However, there has been renewed interest in interpretable algorithms, including decision trees, in the interest of intelligibility (see citation list below). These works argue that for some use cases, interpretable algorithms achieve similar accuracy to state-of-the-art models like neural nets. Our approach complements this line of work because it can prove that the output of an interpretable decision tree algorithm is consistent with respect to a fixed amount of data bias or poisoning. We will clarify this aspect in the paper, along with a disclaimer that decision trees are not suitable for some tasks, marking a limitation of our work.
>
> Citations (we’ll add to the paper):
> * Angelino, E., Larus-Stone, N., Alabi, D., Seltzer, M., and Rudin, C. (2018). Certifiably optimal rule list for categorical data. Journal of Machine Learning Research.
> * Lou, Y., Caruana, R., Gehrke, J., and Hooker, G. (2013). Accurate intelligible models with pairwise interactions. SIGKDD International Conference on Knowledge Discovery and Data Mining (KDD). https://doi.org/10.1145/2487575.2487579
> * Rudin, C., and Radin, J. (2019). Why Are We Using Black Box Models in AI When We Don’t Need To? A Lesson From An Explainable AI Competition. Harvard Data Science Review, 1(2). https://doi.org/10.1162/99608f92.5a8a3a3d

---

> > ### Comment · Reviewer_FQR1 · 2021-08-26
> > **Re: Response**
> >
> > Thanks you for your response. My interpretation was the 2nd which you have adequately answered.
> >
> > I have changed my score appropriately.

---

### Official Review · Reviewer_ov7B · 2021-07-15

**Rating:** 8
**Confidence:** 4

**Summary:**

This paper introduces and explores a framework for certifying that classification models produced by a machine learning algorithm are making predictions on examples that are robust to bias in the training data.  Certification is per example: a certification for an example means that the same prediction would have been made on that example if the training data used to train that model had been perturbed in ways that correspond to potentially eliminating a specified type of bias in the dataset.   The paper presents an algorithm aimed at certifying robustness for a standard decision tree learner and proves that it is sound.  It explores the algorithm in a series of experiments on standard datasets, with different bias definitions and across different demographic groups.

**Limitations And Societal Impact:**

- See comment (9) above.
- In the Checklist at the end of the paper, the authors state as an example that the data split is discussed in the supplemental material.  However, I could not find any discussion in the paper or the supplemental material of how the datasets were divided into train and test.  (Please let me know if I missed it.)  The authors appear to have used a single test set for each dataset, given that they said that error bars are N/A.  This needs to be clarified.

**Main Review:**

The problem addressed in the paper is interesting and important.  The abstract framework and definition of robustness against bias is well-done and provides a valuable theoretical underpinning for future work on bias in training data.   The presented decision tree algorithm demonstrates that this abstract framework can be applied to specific learning algorithms.  I would view the experiments in this paper as preliminary.   As the authors acknowledge, there are important questions raised by the results.  Further experiments and analysis are needed.

The paper is well-written.

Specific comments:

p. 4 In the definition of $MISS_m^g(T)$, the "restrict $T'$" part of the definition uses the notation $.g(x,y)$.   Replace this dot notation with something more standard.

p. 5  The given definition of cost is not normalized, and can be bigger than 1 (as in Example 4.2).  However, in the experimental section, p. 7, the cost of the optimal top-most split is given as 0.30, 0.35, and 0.45, and these seem to be normalized.

p. 5  The text states that argmax returns a set of possible indices.  It should also describe which possible indices are returned.

p. 7 The definition of $infer^a$ is confusing, despite the explanation that "the range of $\cup$ is over predicates of $\Phi^a$.  I believe that the first $\cup$ is over all $\phi$ in $\Phi^a$ such that $\phi(x)$ holds, and similarly for the second one.  Should explain more clearly or change the notation in the definition,

p. 7 The experimental evaluation section talks about "the percentage of test samples" but there is no explanation of how the test samples were chosen.  I did not find this information in the supplementary material either.

p. 9 "Understanding why this phenomenon occurs is beyond the scope of this paper."   I appreciate that the authors performed these experiments on targeted bias models and acknowledged that they do not understand the reasons behind the results.    That said, this phenomenon (or more specifically, these particular experimental results) raises questions about how we should interpret any experimental results obtained using this system, including the other results presented in this paper.   The authors would have done better to explicitly say that the experiments they performed were preliminary, and that further experimentation is needed.

**Time Spent Reviewing:**

3

---

> ### Author Response · Authors · 2021-08-10
> **response to reviewer feedback**
>
> We thank the reviewer for their helpful comments and thorough review!
>
> All of the reviewer’s suggestions on notation or additional explanation make sense to us, and we’ll incorporate them into the paper. Regarding the cost comment, good catch! The numbers on page 7 are normalized by the size of the dataset to allow for head-to-head comparison (we’ll clarify this aspect in-text).
>
> **Train/test split details**: Thanks for bringing this up - we mistakenly omitted details about this in the appendix PDF, but the exact splits are located in the code portion of the supplemental material (see the data folder). We will modify the appendix to include details about the train/test split in the final version. For Adult Income and MNIST-1-7, we used the standard train/test split from the UCI ML repository. For COMPAS and Drug Consumption, there is no default train/test split, so we created our own random split using sklearn’s train_test_split functionality. For COMPAS, we used a 75/25 train/test split and for Drug Consumption we used a 67/33 split.

---

### Official Review · Reviewer_Ycxe · 2021-07-17

**Rating:** 6
**Confidence:** 2

**Summary:**

The paper focuses on robustness against bias in training data. In particular, the main goal of the paper is to develop a method that can certify models produced are pointwise-robust to potential dataset biases. Authors focus on decision-tree learning due to the interpretable nature of the models. Authors propose using a novel symbolic technique to evaluate a decision-tree learner on a large number of datasets, certifying that each and every dataset produces the same prediction for a specific test point.

**Main Review:**

General Assessment: Paper tackles a relevant problem, is well written and empirical evaluation is sound. Authors consider various models of biases and then develop methods to certify robustness on each of those.

Detailed Review:
- Authors mention that their method certifies robustness but not accuracy. It would be interesting if the method can be extended to certify accuracy.
- Authors currently need users to identify the assumptions underlying the biases. It would be interesting to see how this method might extend beyond this requirement.

**Time Spent Reviewing:**

2

---

> ### Author Response · Authors · 2021-08-10
> **response to reviewer feedback**
>
> We thank the reviewer for their thoughtful review! To address the reviewer’s points from the “detailed review” section:
>
> **Accuracy:** For the test data (where we have a ground-truth label to compare to), we can easily extend our work to certify accuracy by comparing the decision trees’ output to the ground truth. At low levels of bias, the accuracy of certified test instances is similar to the overall accuracy. However, as the bias increases (and we’re able to certify fewer elements overall), the accuracy of the certified elements improves. For example, for 0.4% bias for Adult Income, the 34.8% certification rate has 99% accuracy. And on the COMPAS dataset, where the overall accuracy is just 64%, a bias of 0.4% yields a 9.3% certification rate with 81% accuracy.
>
> **Users identifying bias assumptions:** This is an interesting comment, and something we haven’t thought of before! Are you envisioning automatically detecting a data collection flaw from the data itself? Our initial thought is that expert input would still be required because even if we automatically detected a discrepancy in the data about two demographic groups, we would not know whether that is a data flaw or a valid difference between two groups without a human to explore potential causality paths. However, maybe there’s something we haven’t thought of. In any case, this seems like an interesting direction for future research.

---

> > ### Comment · Reviewer_Ycxe · 2021-08-24
> > **Response**
> >
> > Dear Authors,
> >
> > Thanks for your responses. I have also read the other reviews for this paper and I still stand by this paper.
> >
> > Regards,
> > Reviewer Ycxe

---

### Official Review · Reviewer_RG8X · 2021-07-18

**Rating:** 6
**Confidence:** 4

**Summary:**

The paper proposes to certify that models are pointwise-robust to potential dataset biases using decision trees. The approach programmatically specifies bias models), i.e. models that specify suspected bias. Robustness is certified using a symbolic technique to evaluate a decision tree on a large number of datasets and certifying that all datasets provide the same decision for a test point.

**Limitations And Societal Impact:**

The authors explained about their choice of the datasets they used; I don't have concerns here.

**Main Review:**

The concept of bias models as proposed in this paper is interesting, but I would perhaps name it differently because ‘bias model’ is quite an overloaded word. The paper breaks down bias models into a few types: missing data, label flipping, fake data. Additionally the paper suggests targeted versions for each of the three types where only a specific group of data points experience that bias, and composite versions which are arbitrary combinations of missing, label, and/or fake. It may be good to state that these are not exhaustive and there are more potential issues that could lead to all datasets not providing the same decision for a test point. E.g. besides label flipping, what about feature flipping which could lead to a change in the label?

Evaluation:
-	Can you do an approximate baseline by enumerating some datasets in the bias set, or any other baseline you can provide? It is hard to contextualize the results otherwise.
-	Can you give an idea of how long the procedure takes?
-	I’m a bit perplexed by the targeted bias results for COMPAS, where at first glance FLIP^g and FLIP^{g’} seem like they should be two sides of the same coin. For COMPAS with n=4629 to confirm, did you limit it to only white and black? What did the tree look like?

**Time Spent Reviewing:**

1

---

> ### Author Response · Authors · 2021-08-10
> **response to reviewer feedback**
>
> We thank the reviewer for their thoughtful review! We will incorporate the reviewer’s suggestion to mention the non-exhaustiveness of the bias models as a limitation. In response to other questions/comments:
>
> **“Bias model” terminology** - This is a fair point. We originally called them “poisoning models” but switched to the “bias” phrasing to encompass situations where the data was not maliciously modified (such as bias that occurs due to uneven sampling procedures). We can switch back to saying “poisoning models” (we welcome alternate suggestions from the reviewer, though).
>
> **Feature flipping** - We did look at feature flipping and decided that under this framework, it’s not too interesting from a technical perspective. (If Feature X may be poisoned, then it essentially boils down to excluding Feature X from the model. To see this, think about what happens if we split on the poisoned feature: we don’t know which data were poisoned, so we must include all data in both branches of the tree.) On the other hand, restricting feature flipping to a targeted domain is a more interesting problem, but one that will have to wait for future work.
>
> **Baseline** - We explored this question for the COMPAS dataset by randomly perturbing training data (within a certain bias threshold) and then building decision trees to test whether each specific perturbation broke robustness (i.e., gave a different prediction for a fixed input than the original classifier did). Details about this procedure and its results are in Appendix E.4.
>
> **Procedure runtime** - The amount of time required to certify a single input depends on the dataset and the bias model. As explained in Appendix E.2, datasets with larger and more complex feature spaces require more time, as do bias models with a larger absolute level of bias. We already include some sample runtimes in the Appendix, but we will add a standalone table containing more details. Notably, for the COMPAS and Drug Consumption datasets using depth-1 trees, all bias models we tried required less than 0.1 seconds of runtime. For the Adult Income dataset, using the FLIP bias model and depth-2 trees, the tests with 0.1% bias or less all required around 0.5 seconds. But increasing the bias beyond 0.1% incurs a sharp increase in runtime: 0.2% bias took about 75 seconds, 0.4% bias about 13 minutes, and 0.6% bias about 90 minutes.
>
> **Difference between FLIP$^g$ and FLIP$^{g’}$** - For data points with races other than Black or White, we include them in the dataset, but do not modify them under either of the targeted bias models. We will edit the paper to clarify this.
>
> * What did the tree look like? -  When there is no poisoning, the tree splits on ‘Number of prior offenses $\leq$ 1’. (COMPAS experiments have depth 1, so really it’s a stump, not a tree.)
>
>   * For g = (race=Black and label=positive), at a small (0.2%) level of bias, optimal trees all split on ‘Number of prior offenses’, but the cutoff point may be 1, 2, 3 or 4. For 0.5% label-flipping bias, the additional splits on the number of prior offenses $\leq$ 5, 6 or 7 are added. For 1% bias, the optimal trees include all the above ones, plus splitting on prior offenses $\leq$ 8, 9, 10, 11, 12, and 13, as well as on ‘age above 45,’ ‘age below 25,’ ‘Hispanic,’ ‘Female,’ and ‘Misdemeanor.’
>
>   * For g = (race=White and label=negative), the results are similar. The main difference is that the cutoff for ‘number of prior offenses’ has a slightly larger range at each level (the cutoffs range from 1-5 for 0.2% bias, 0-9 for 0.5% bias, and 0-18 for 1% bias).

---

### Decision · Program_Chairs · 2021-09-27

**Decision:**

Accept (Poster)

**Comment:**

The paper proposes to certify that machine learning models (in particular decision trees learned by the CART algorithm) are robust to potential dataset biases. This is a timely and important topic, given the widespread deployment of ML models in high stakes situations, and the difficulty of collecting data that are fully representative and free from biases.

Reviewers were in consensus that the paper makes an interesting contribution to the literature in this important problem space. Only minor concerns were raised in the initial reviews, and adequately addressed by the authors during the rebuttal.

I recommend acceptance and congratulate the authors on their timely and important research work.